# Synergizing habits and goals with variational Bayes

Dongqi Han [1] ✉, Kenji Doya [2], Dongsheng Li [1] & Jun Tani [2]

Behaving efficiently and flexibly is crucial for biological and artificial embodied agents. Behavior is generally classified into two types: habitual (fast but inflexible), and goal-directed (flexible but slow). While these two types of behaviors are typically considered to be managed by two distinct systems in the brain, recent studies have revealed a more sophisticated interplay between them. We introduce a theoretical framework using variational Bayesian theory, incorporating a Bayesian intention variable. Habitual behavior depends on the prior distribution of intention, computed from sensory context without goal-specification. In contrast, goal-directed behavior relies on the goal-conditioned posterior distribution of intention, inferred through variational free energy minimization. Assuming that an agent behaves using a synergized intention, our simulations in vision-based sensorimotor tasks explain the key properties of their interaction as observed in experiments. Our work suggests a fresh perspective on the neural mechanisms of habits and goals, shedding light on future research in decision making.

In psychology and cognitive neuroscience, intelligent agents, such as humans and other mammals, are thought to engage in two types of behavior: habitual and goal-directed[1–3]. Habitual behavior refers to the actions that are performed automatically, without conscious thought or intention, in order to maximize the agent's benefits (rewards), such as seeking food and avoiding danger. Habitual behavior is typically computed in a model-free manner, i.e., it does not require the agent to consider the detailed consequences of their actions. On the other hand, goal-directed behavior describes the actions that are performed with the aim of achieving a specific goal, such as going to a certain place. Goal-directed behavior is based on knowledge about the world (model-based), and it is typically more flexible and responsive to changes in the environment, as it involves conscious decision-making and planning using an environment model[4,5]. A well-known phenomenon is the automatic transition from goal-directed to habitual behavior with repetitive training[6,7], which is not yet fully understood.

The two types of behaviors have also been extensively studied in machine learning and deep learning, especially in decision-making and control problems[8]. Reinforcement learning (RL)[9] is a computational paradigm that considers learning a policy (that is, the strategy to choose actions) that maximizes rewards. Model-free RL (MFRL), which

does not involve an environmental model, is particularly well aligned with the acquisition of habitual behavior[10]. On the other hand, goal-directed behavior is well captured by active inference (AIf) theory[11,12]. The AIf theory, a computational framework in cognitive science and neuroscience, posits that agents strive to minimize surprise or uncertainty in their interactions with the environment. The core concept in AIf, free energy[13], quantifies the divergence between an agent's probabilistic predictions of sensory input and the desired sensory inputs (in the contexts of pursuing a goal[14]). Minimizing this difference gives rise to the agent's goal-directed behavior. To clarify, we distinguish goal and reward: reward is a scalar value representing the benefits to the agent that are to be maximized, whereas goal refers to the state (e.g., position or visual perception) that is to be achieved.

Conventionally, habitual and goal-directed behaviors have been treated independently of each other in both cognitive science and machine learning. Although there are behavioral studies considering a hybrid schema to explain animal or human behavior, the most common way is to simply model an agent's estimated value for its action as a linear combination of the habitual and goal-directed ones[4,5,15]. In machine learning, one practical reason for such a separation is that the inputs are different − in goal-conditioned control or decision-

[1]Microsoft Research Asia, Shanghai 200232, China. [2]Okinawa Institute of Science and Technology, Okinawa 904-0495, Japan.
✉ e-mail: dongqihan@microsoft.com

making[16], the goal is usually an input to the model. Thus, the model for goal-directed behavior has additional input of the goal compared to that for habitual behavior. Therefore, it is common for two separate models to be designed for these two behaviors when both are considered[17,18].

However, we argue that these two systems should be more closely related to each other and should be learned and utilized synergistically. Although it has not been fully understood how interactions between these two systems occur in the brain[2–5,19–22], habitual and goal-directed behaviors share common downstream neural pathways such as the brainstem[19]. The conjecture is that habitual and goal-directed behaviors share the low-level motor skills, therefore, each system may leverage the well-developed actions learned by the other. The question then arises: How can we realize a synergy of both behaviors while considering their differences? Recent works have attempted to answer this question. Matsumoto et al.[23] introduce a variational Bayesian neural network model which generalizes the movement of the habituated robot arm to the goal-directed ones. However, it requires supervised data on detailed trajectories of robot joint angle and makes a strong assumption that it is known how to determine the joint torque in a specific end-effector position and orientation (inverse kinematics), which is often impractical in the real world[24,25]. Moreover, Friston et al.[20] and Schwöbel et al.[21] propose to model both kinds of behaviors within the AIf framework. However, their work did not elucidate the role of RL in behavior learning[10] and did not involve sensorimotor learning (see more discussion in Supplementary Related Work), leaving a gap that further research needs to address.

In this research, we propose a theoretical framework for understanding behavior in sensorimotor learning based on variational Bayesian methods[26]. This framework, which we term Bayesian behavior, is grounded in deep learning principles and centered on a probabilistic latent variable $z$ that we refer to as the intention of an agent. We conceptualize intentions for habitual and goal-directed behaviors as the prior and posterior distributions of $z$, respectively. In statistics, prior distributions are probability distributions that represent beliefs about a quantity before having certain data, whereas posterior distributions are probability distributions that represent beliefs about the quantity with the given data. In our framework, the goal-directed intention is additionally conditioned on the goal than the habitual one, and we have

$$\text{habitual action} \leftarrow z^{\text{prior}}(\text{sensory observation})$$
$$\text{goal-directed action} \leftarrow z^{\text{post}}(\text{sensory observation, goal})$$

where $\leftarrow$ denotes the neural pathway (policy function) to generate action from $z$. The policy function is shared for both habitual and goal-directed actions, and can be trained by MFRL using the agent's experiences[10]. The prior distribution of $z$ is not conditional on the goal but depends solely on contextual cues[3]. In contrast, the posterior distribution incorporates a goal to achieve in the future. This additional conditioning differentiates the posterior distribution from the prior distribution, allowing for goal-directed behavior. As habitual and goal-directed intentions are probabilistic variables, our framework naturally accounts for the arbitration between habitual and goal-directed behaviors based on their uncertainties[6], which can be directly given by the statistical variance of $z^{\text{prior}}$ and $z^{\text{post}}$.

Our framework includes minimal artificial assumptions in learning: the model only performs free energy minimization[13] and MFRL with experience replay[10]. Nevertheless, we will demonstrate the aptness of the Bayesian behavior framework to address the following critical questions[2,3]: (1) How does an agent arbitrate between model-free, habitual behavior and model-based, goal-directed one? (2) How does an agent autonomously transfer from slow, goal-directed, to fast, habitual behavior with repetitive trials? (3) How does an agent perform

goal-directed planning for a novel goal that has not been trained to accomplish?

We aim to explore these questions through simulated experiments using a robot agent. Moving beyond prior computational models that predominantly focused on discrete states and actions[5,15,21], our approach is empowered by deep learning to simulate a first-person vision-based robot that interacts with an environment using continuous motor actions. In particular, we show the dynamic change of the properties of prior and posterior intentions during the learning and behaving process, which affects the arbitration between them. Furthermore, our findings offer fresh perspectives on the neural mechanisms that underpin the cost-benefit trade-offs in the behaviors of intelligent biological agents[27]. We illustrate how RL and AIf, two paradigms traditionally seen as opposing[28], can work in synergy to foster efficient and flexible behavior. In summary, our results provide both a theoretical and computational narrative that elucidates the interplay between habits and goals in behavior, shedding light on decision-making research in both neuroscience and the burgeoning field of embodied AI.

## Results

### Sensorimotor learning and behaving scheme

We consider an agent that dynamically interacts with an environment (see Fig. 1a as an overview). The agent receives sensory observations and reward signals from the environment, which are used to train its internal world model (for sensory prediction) and action model (or policy in RL terminology[9]). To behave, the agent arbitrates between goal-directed and habitual intentions, and then it executes actions based on the intention to interact with the environment (Fig. 1c). The agent's experiences of interacting with the world are memorized for learning (Fig. 1b), a process also known as experience replay[29]. We consider the case in which an agent performs sensorimotor learning for navigation tasks tabula rasa, i.e., through self-exploration.

### Introducing the Bayesian behavior framework

The proposed Bayesian behavior framework is centered on a Bayesian latent variable $z_t$, referred to as intention (Fig. 1). The subscript $t$ denotes the timestep as we consider a Markov decision process[8]. We use the superscript letter $p$ to denote the prior of intention ($z_t^p$) and $q$ for the posterior ($z_t^q$). The intention $z_t^p$ (or $z_t^q$) is a random variable following diagonal Gaussian distribution $\mathcal{N}(\boldsymbol{\mu}_t^p, \boldsymbol{\sigma}_t^p)$ (or $\mathcal{N}(\boldsymbol{\mu}_t^q, \boldsymbol{\sigma}_t^q)$). In this work, the intention is a four-dimensional vector probabilistic variable, encoding the agent's behavior tendency and prediction about observations. Such a compact latent representation is consistent with the idea of predictive coding[30,31] (PC), a theoretical framework suggesting that the brain uses an internal generative model to make predictions about incoming sensory information, learning the statistical patterns in the world, and only transmitting unpredictable elements.

The latent intention $z_t$ acts as a nexus to connect motor action and sensory prediction. Our framework adaptively regulates the habitual and goal-directed intentions by minimizing the variational free energy (mathematically equivalent to the negative variational lower bound in variational Bayesian methods[26], see Supplementary Proof for mathematical derivation):

$$\text{To minimize : free energy} = \underbrace{\text{observation prediction error}(z_t^q)}_{\text{accuracy}}$$
$$+ \underbrace{\text{KL-divergence}(q(z_t) \| p(z_t))}_{\text{complexity}} \quad (1)$$

The first term reflects the basic idea of PC, which involves learning an internal model about the environment (Fig. 1b,c). The internal model predicts current and future sensory observations given the agent's intention. The second term is the Kullback–Leibler divergence[32] (KL-

divergence, a measurement of how different two probabilistic distributions are) between the posterior and prior distributions of intention. Intuitively, the KL-divergence term balances the fit of the model to the data with the complexity of the latent intention $z_t$. We highlight the KL-divergence (complexity) term as the key to synergizing habitual and goal-directed behaviors by bounding their difference. This concept is known as information bottleneck in machine learning, to which we also welcome the readers to refer[33,34] (see Supplementary Discussion). The following sections will further elucidate the pivotal roles of the complexity term in learning and behaving.

**Learning and behaving under the Bayesian behavior framework**
Our Bayesian behavior framework contains several parts. We explain the algorithm and model during learning (training the model using experience replay) and behaving (interacting with the environment), respectively (Fig. 1b, c).

In learning (Fig. 1b), there is a habitual system that accounts for the generation of habitual intention, including an encoder to receive visual observations and a recurrent neural network (RNN) to process contextual information. The habitual, prior intention $z_t^p$ is calculated in a straightforward way using contextual observations, in line with the

well-recognized idea in cognitive science and psychology that habitual behavior conditions context signals while ignoring future outcome[3,35]. Another part, the goal-directed system, is trained to predict and encode current and future observations (since future observations are available in experience replay) (Fig. 1b), similar to an auto-encoder[36]. This goal-directed system reflects the idea of PC for current observation[30,37] and future observation (also known as temporal PC[38,39]) since the goal-directed action should depend on both the current context and the desired future outcome. The free energy[13] being minimized in learning is:

$$\mathcal{F} = \underbrace{- \ln P(\text{pred. current obs. by } z_t^q \text{ is correct}) - \ln P(\text{pred. future obs. by } z_t^q \text{ is correct})}_{\text{prediction errors}}$$
$$+ \underbrace{D_{\mathrm{KL}}\left[q(z_t) \| p(z_t)\right]}_{\text{complexity}}$$

$$(2)$$

where $\ln P$ is the log-likelihood, a higher value of which means a better prediction. KL-divergence is denoted by $D_{\mathrm{KL}}$, and $q(z_t)$, $p(z_t)$ are the probability density function (PDF) for posterior and prior intentions, respectively.

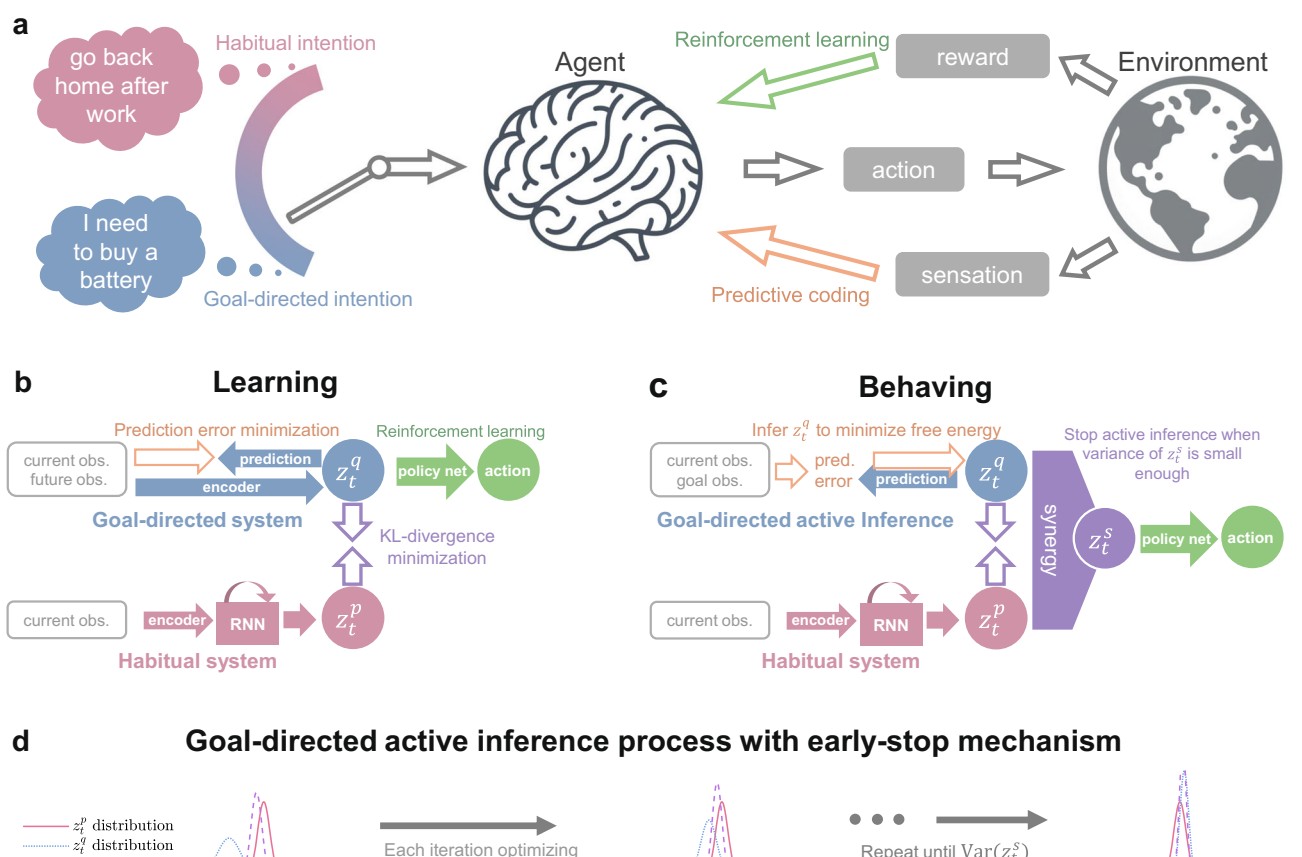

**Fig. 1 | The Bayesian behavior framework. a** Overview of the behavioral scheme considered in this work. The agent learns to select the optimal action through reinforcement learning using reward signals and performs predictive coding using sensory observations (learning the world models). **b** Diagram of our framework during training. The entire model is optimized to minimize RL loss and free energy (Eq. (8)). The model's synaptic weights and biases are updated end-to-end through gradient descent, utilizing memorized experiences. Note that the reward signals affect all networks in learning through backpropagation. **c** Diagram of our framework during behaving. The goal-directed intention $z_t^q$ is obtained by active inference with respect to the goal (sampled from the agent's memorized observations at the positions with maximal reward). The action of the agent is determined by the synergized intention $z_t^s$, calculated as the inverse variance weighted average of the two intentions. **d** Iterative process to minimize the corresponding free energy by optimizing the probability density function (PDF) of the posterior intention (i.e., active inference). The active inference process may stop early if the mean variance of $z_t^s$ is small enough to save computation cost. For clarity, a one-dimensional intention is visualized here.

The policy network, shared for both habitual and goal-directed behaviors, is trained by RL using the soft actor-critic algorithm[40]. The total loss function in learning (Fig. 1b) simply adds up free energy and the loss of the RL algorithm[40] (Eq. (8)). All network models are trained end-to-end using stochastic gradient descent and experience replay, and it is continuously being updated during the learning process. There is also a value function model[40] to estimate the reward value of the policy, trained in parallel, which is not shown here for clarity, as its gradients in learning do not affect the main model (see Methods). Note that although the action is computed using posterior intention in learning (Fig. 1b), the habitual behavior is also optimized considering KL-divergence term according to variational Bayesian theory (see Methods Eq. (12) for mathematical explanation). In other words, the prior, habitual behavior and posterior, goal-directed behavior are learned in synergy via RL and KL-divergence minimization.

Now, we explain our framework when the agent is behaving (Fig. 1c). The habitual system works in the same way as learning, computing a prior intention $z_t^p$ in a straightforward way, which is fast, since it only needs one-time inference. Meanwhile, for the goal-directed system, as we consider a vision-based sensorimotor learning scheme, the goal is an image indicating the desired outcome (e.g., goal observation can be a shelf with batteries when the goal is buying a battery). Here, we assume a simple principle to choose the goal: randomly sampling from the agent's memorized observations at the best-rewarded positions, reflecting the idea that the goal should be value-sensitive[3]. As shown in Fig. 1c, the goal-directed system computes $z_t^q$ via AIf. More specifically, the free energy w.r.t. the goal can be written as follows.

$$\mathcal{L}^{\text{AIf}} = \underbrace{-\ln P(\text{pred. current obs. by } z_t^q \text{ is correct}) - \ln P(\text{pred. future obs. by } z_t^q = \text{goal obs.})}_{\text{prediction errors}}$$
$$+ \underbrace{D_{\text{KL}}\left[q(z_t)||p(z_t)\right]}_{\text{complexity}}$$

$$(3)$$

This free energy w.r.t. goal is similar to that during learning (Eq. (2)), while the difference is that in AIf the predicted future observation should be close to the goal observation. Technically, the AIf process is realized by iterative optimizations of the distribution of $z_t^q$ (Fig. 1d, see Methods for details). While here we use evolutionary strategy[41,42] for computational efficiency since $z_t^q$ is low-dimensional, gradient descent is also a way[43]. Nevertheless, this model-based mental search process (or optimization problem in machine learning) is essential to realize planning for untrained goals[43,44], consistent with the scientific recognition that goal-directed behavior is slow to generate and demands more computational efforts[2,6].

Having both $z_t^p$ and $z_t^q$, a natural assumption is that the agent pursues a lower variance of its actual behavior output by leveraging both intentions. This can be achieved by inverse variance weighted average, a well-known method in statistics that minimizes the variance of the combined value of two random variables:

$$z_t^s = \frac{w_t^p z_t^p + w_t^q z_t^q}{w_t^p + w_t^q}, \text{where } w_t^p = (\sigma_t^p)^{-2}, \text{and } w_t^q = (\sigma_t^q)^{-2} \quad (4)$$

The superscript letter $s$ stands for the word synergized, and we refer to this inverse variance weighted average of $z_t^p$ and $z_t^q$ as the synergized intention $z_t^s$. The multiplications of two vectors are element-wise in this article, if not specified. Our framework proposes that the agent utilizes the synergized intention for acting (Fig. 1c) in order to achieve the optimal precision. Such a synergized intention also yields a competition between $z_t^p$ and $z_t^q$, where the one with lower variance (($\sigma_t^p)^2$ or ($\sigma_t^q)^2$) will dominate (with larger weight). Interestingly, this idea is also consistent with a popular hypothesized substrate underlying the arbitration between habitual and goal-directed behaviors[5,6,45], where

the one with lower uncertainty (thus higher confidence) is preferred. Elegantly, the uncertainty of habitual or goal-directed behavior is straightforwardly represented by the variance of the corresponding intention $z_t^p$ or $z_t^q$ (since they are Gaussian random variables) in our framework. Another view can be drawn from the equality that $\text{PDF}(z_t^s) = \text{PDF}(z_t^q) \cdot \text{PDF}(z_t^p)$, which interprets the synergy as a Bayes rule[21].

Last but not least, as the goal-directed AIf is computationally costly, we introduce a mechanism to save cost in computing the synergized intention (Fig. 1d). It can be easily computed that $\sigma_t^s$, the standard deviation (STD), i.e. square root variance, of the synergized intention, satisfies (for all dimensions of the intention)

$$(\sigma_t^s)^{-2} = (\sigma_t^p)^{-2} + (\sigma_t^q)^{-2} > (\sigma_t^p)^{-2}. \quad (5)$$

If the prior precision (inverse variance $(\sigma_t^p)^{-2}$) is high, indicating the agent is confident about its habitual behavior, then $(\sigma_t^s)^{-2}$ will also be high (thus $\sigma_t^s$ is low). Since the agent aims for a low variance of its synergized intention, the AIf process (Fig. 1d) may reduce the number of iterations if $\sigma_t^s$ is sufficiently low. In our framework, we terminate the AIf process if $\text{mean}(\sigma_t^s) \leq \sigma^{\text{thres}}$ computed from $\sigma_t^q$ in any iteration during AIf (see Methods). In the extreme case that $\text{mean}(\sigma_t^p) \leq \sigma^{\text{thres}}$, i.e. the habitual intention alone is precise enough, AIf only needs one iteration, thus spending minimal cost. This mechanism explains the phenomenon that a person is less likely to perform outcome-oriented planning (e.g., scheduling a route with a map) when habitual behavior (e.g. going home from work) becomes more proficient. In addition, one might find that $\sigma^s$ is not directly affected by $\mu_t^p$ and $\mu_t^q$. This seems contradictory to the intuition that the uncertainty of $z_t^s$ should be large when $|\mu_t^p - \mu_t^q|$ is larger. However, minimizing the KL-divergence leads to an increase of $\sigma_t^p$ and $\sigma_t^q$ when $|\mu_t^p - \mu_t^q|$ is large (Eq. (10)), thus the difference between $\mu_t^p$ and $\mu_t^q$ indirectly affects $\sigma^s$.

To summarize, there are two basic assumptions of our proposal: first, habitual and goal-directed intention as the prior and posterior of a variational Bayesian variable, constrained by a KL-divergence between them in both learning and behaving; and second, the agent behaves based on the synergized intention that is automatically computed by combining habitual and goal-directed intentions. Unlike the conventional view of treating habitual and goal-directed systems separately in learning[5,6], our framework enables reward-maximizing motor skills by RL and goal-achieving reasoning ability by AIf to be shared in both systems through the bridge of the KL-divergence term in free energy (the gradients of training losses communicate bidirectionally through the KL-divergence term in learning).

The following sections will show and explain the simulation results capturing the key properties of habitual and goal-directed behaviors. First, we explain how an agent automatically transitions from computationally heavy, goal-directed behavior to rapidly computed, habitual behavior gradually with repetitive trials (Fig. 2). Then, we analyze the change in the agent's behavior after a reward reconfiguration (outcome devaluation[45]), showing more immediate change and faster re-adaptation of the behavior of agents with moderate training than those with extensive training. Finally, we demonstrate that the trained agent can flexibly perform goal-directed planning for novel goals.

## Vision-based T-maze environment
We focus on a relatively simple, yet important navigation environment in a T-shaped maze, or simply T-maze (Fig. 2a). The T-maze environment is a common behavioral paradigm used in cognitive science to study learning, memory, and decision-making processes in animals[46,47]. Here, we consider a variant of the T-maze, in which the objective of habitual behavior is to escape from the maze as soon as possible, assuming that an enemy is chasing the agent. There are two exits in the top-left and top-right corners (Fig. 2a). In each trial (episode), the agent

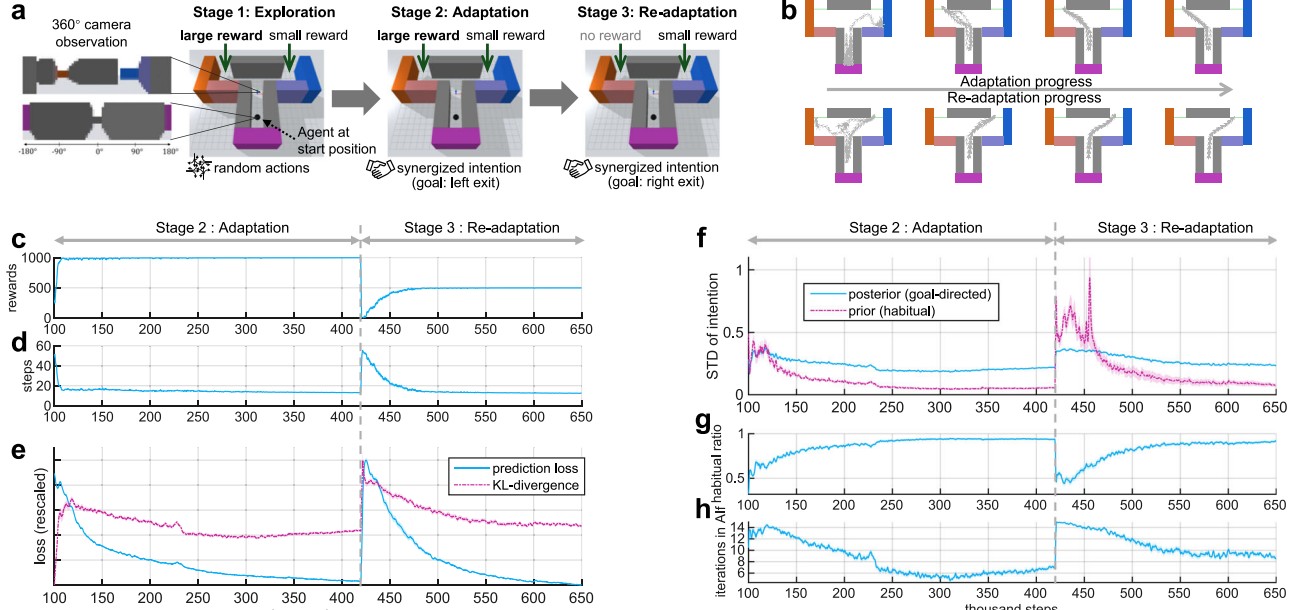

**Fig. 2 | Simulation results of the habitization experiment. a** Explanation of the environment and task. The agent performs random actions in the first exploration stage to collect experiences (no learning). During stages 2 and 3, the agent keeps learning and follows its synergized intention. The task is episodic, meaning that the agent will be set to the initial position in each trial. **b** An example agent's behavior (movement trajectories of 5 trials in each plot, aerial view) throughout the experiments. It first learns to achieve the left exit where larger reward is given in stage 2 (first row) and then re-learns to go to the right exit when the left reward is extinguished in stage 3 (second row). **c** Episodic rewards during stages 2 and 3. **d** Steps taken in one trial to reach exit (a trial is forced to terminate if it reaches 60 steps). **e** The values of prediction error and KL-divergence in free energy loss during training process (see Supplementary Fig. 1 for a discussion of the balance between them). **f** Standard deviation (STD) of prior and posterior intentions in behaving. Here, the plotted posterior distribution is obtained by full goal-directed AIf process without early stop (independent of the actual posterior used). **g** The habitual ratio computed by mean$(\sigma_t^p)^{-2}/((\text{mean}(\sigma_t^p)^{-2}+(\text{mean}(\sigma_t^q)^{-2})$. A higher habitual ratio indicates that the habitual (prior) intention is more dominant in behavior. **h** Number of iterations in AIf needed to make the variance of the synergized intention small enough. Fewer iterations means faster action computation. For **c**–**h**, the horizontal axes denote environmental steps in stages 2 and 3, where the first 100 thousand steps not shown here are for exploration (stage 1). Curves indicates the mean result of 50 random seeds, and the shaded areas of the curves indicate standard error of the mean. Curves are smoothed for clarity.

starts from a fixed initial position at the bottom (Fig. 2a) of the maze. If the agent reaches an exit, it will receive a reward, and this trial is finished. Hitting the wall once will result in a small amount of negative reward for the agent. To make the environment more realistic like that of a biological agent, the observation of the agent is visual perception − a 360° RGB camera centered on the agent with a resolution of 16 by 64 (Fig. 2a). We consider continuous-valued motor actions: the agent can decide its horizontal movement at each step (represented by a two-dimensional action vector) with a speed limit. We consider a discount factor[9] of 0.9 for RL so that it is encouraged to escape with fewer steps.

### Automatically transition from goal-directed to habitual behavior

It is well known that when an animal learns to solve a new task, it first relies mainly on goal-directed systems, and then the behavior transitions to become more habitual and faster with repetitive trials[3,35,48]. However, the underlying mechanisms are not yet fully understood. In this section, we use the Bayesian behavior framework to explain how an agent's behavior intention dynamically changes with learning in the T-maze environment. Here, we consider two cases: learning from scratch and with an existing habit (reward devaluation). The simulated experiment includes three consecutive stages (Fig. 2a): an exploration stage where the agent performs random actions to collect sensorimotor experience for learning later[49]; an adaptation stage where the agent starts learning and behaving (Fig. 1b, c) to pursue maximal reward (1000) in the left exit; and a re-adaptation stage without interrupting learning, but the reward configuration is adjusted so that the optimal behavior is to go to the right exit (reward 500). From stage

2 to 3, the only change is the reward at the left exit becomes zero (agent also updates its memorized reward of left exit) so that the agent should re-adapt its behavior. We adopt a simple method for goal selection: goal observation in each trial is randomly sampled from the agent's memory, where there is a maximal reward (left exit in stage 2 and right exit in stage 3). During stages 2 and 3, the agent's model is continuously trained end-to-end using its recent experiences.

Figure 2b shows the motor trajectories of the agent in the progress of adaptation. Under our framework (Fig. 1b, c), the agent's behavior first converges towards the left exit in stage 2, then re-adapts to the right following the reset of the reward configuration in stage 3. The average episodic reward with respect to environmental steps is shown in Fig. 2c, and the average steps taken to complete each trial (with a forced termination after 60 steps) is shown in Fig. 2d. The KL-divergence and prediction errors in the free energy loss in learning can be seen from Fig. 2e. While these results show that the agent can successfully adapt and re-adapt in this sensorimotor navigation task, our main interest is in understanding how the agent arbitrates between habitual and goal-directed behaviors. This can be explained by the uncertainty-based competition between prior and posterior intentions (Eq. (4)). Figure 2f plots the changing progress of the standard deviation (root-square variance) of the prior ($\sigma_t^p$) and posterior ($\sigma_t^q$) intentions. To quantify the degree to which the agent prefers habitual intention, we define habitual ratio, given by mean$(\sigma_t^p)^{-2}/((\text{mean}(\sigma_t^p)^{-2}+(\text{mean}(\sigma_t^q)^{-2})$, which is the weight of the prior intention in the synergized intention (Eq. (4)). A higher habitual ratio means that the variance of the prior intention is lower compared to the posterior variance; therefore, the agent places greater reliance on the habitual intention $z^p$. It can be seen from Fig. 1g that the habitual

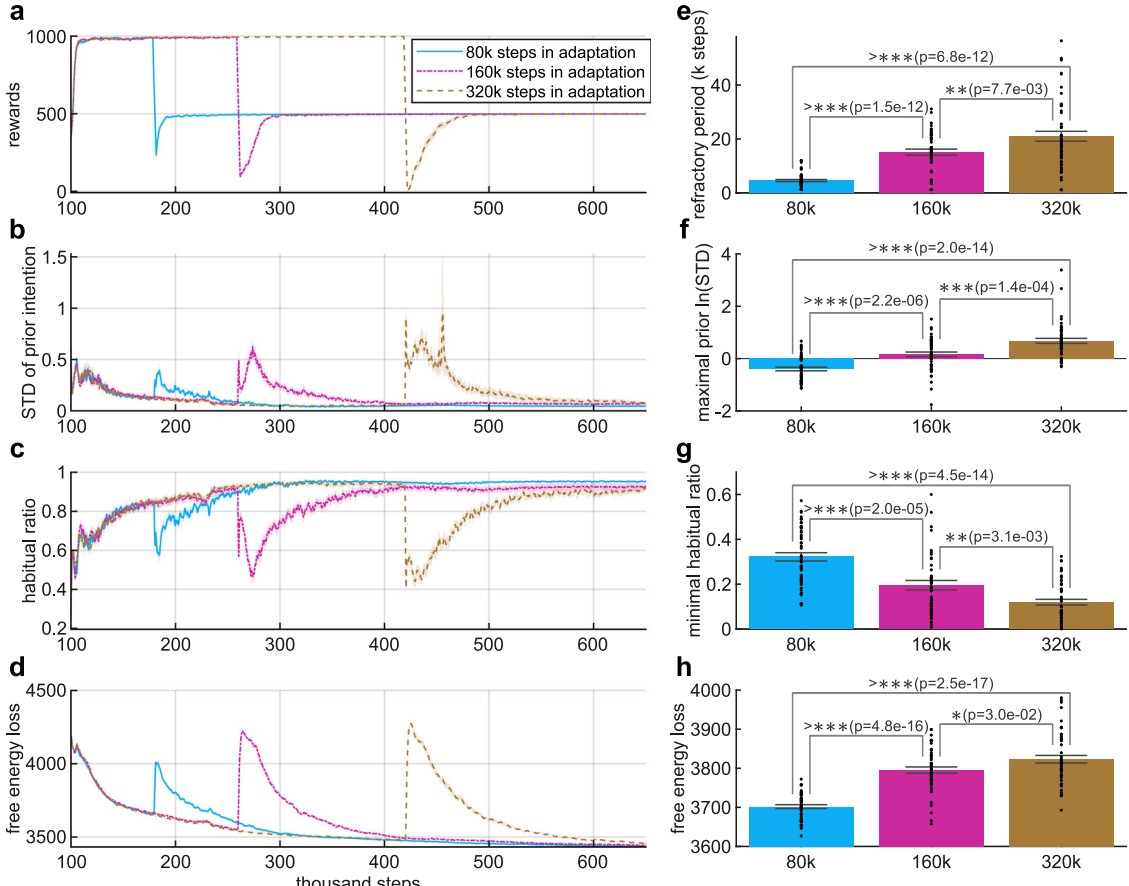

**Fig. 3 | More steps of training in stage 2 make it harder to re-adapt.** Same experimental setting as illustrated in Fig. 2a, but different steps during habitization to the left exit in stage 2 (80k, 160k, 320k steps, respectively). **a**–**d** change of rewards, STD of prior intention, habitual ratio and average free energy loss during the adaptation-re-adaptation process, plotted in the same way as in Fig. 2. Curves are smoothed for clarity. **e** Steps (in thousand) needed to change to re-adapt its behavior for new reward configuration (reaching 80% optimal performance). **f** Maximal natural logarithm of STD of the prior intention in re-adaptation stage. **g** Minimal habitual ratio in re-adaptation stage. **h** Average free energy loss in the first 100k steps of re-adaptation stage. Data are presented as mean value ± standard error of the mean, and two-sided Welch's $t$-test is used for statistical comparison ($n = 50$ random seeds for each case).

ratio gradually increases in stage 2 and similarly in stage 3. This is due to the result that the prior intention becomes more aggressively precise ($\sigma_t^p$ drops faster) than the posterior one (Fig. 2f). An intuitive explanation for this is that as motor actions become more certain with training (Fig. 2b), it becomes easier and likely more accurate to compute behavior in a model-free manner using immediate context cues than by using the more complex AIf process (Fig. 1b, c).

Furthermore, as the variance of the habitual intention gradually decreases during training, it also affects the variance of the synergized intention $\sigma_t^s = \sqrt{1/((\sigma_t^p)^{-2} + (\sigma_t^q)^{-2})}$. According to the mechanism we have introduced to early-stop AIf iterations if the mean of $\sigma_t^s$ is adequately small (Fig. 1d), the number of iterations needed in behaving will be decreasing during the training course in stages 2 and 3, respectively. This conjecture is empirically validated in Fig. 1g, accounting for the fact that behavior becomes faster and computationally cheaper with repetitive training[50,51]. We have also investigated the cases with different precision threshold $\sigma^{thres}$, where similar results are observed (Supplementary Figs. 2–4). Future work could investigate the neuroscientific implications of this early termination mechanism in the brain.

## Extensive training prevents re-adaptation

A well-known phenomenon is that after a change in reward outcomes, extensively-trained behavior becomes more resistant to change[3]. In particular, in the T-maze environment we used, the reward at the left exit is larger in stage 2, but we consider the case that it becomes zero in stage 3. This is analogous to the experimental paradigm known as outcome devaluation[45] If the agent has formed a stronger habit to go to the left, it will be more resistant to re-adapt its behavior to going to the right exit. Here, we use simulated experiments to computationally explain this phenomenon in two aspects: long-term and short-term effects. The long-term aspect is about how the training duration before outcome devaluation (stage 2) affects the agent's re-adaptation process after devaluation (stage 3). On the other hand, we examine the short-term effect by looking at the trials immediately after outcome devaluation, as generally considered in psychological experiments[6,45,52].

First, we investigate the long-term effect of outcome devaluation. We conducted simulated experiments with different durations of stage 2 in the habitization experiment (Fig. 2a). The results are shown and compared in Fig. 3, which includes 3 cases: 80k, 160k, 320k steps in stage 2 (left reward is larger), respectively. It is observed that the refractory period (the environmental steps taken to change behavior) at the beginning of re-adaptation becomes longer with longer training in stage 2 (Fig. 3a, e). In other words, extensive training in stage 2 causes the agent to expend more effort in re-adaptation.

To understand the underlying mechanisms, we then visualize the curves of STD (square root variance) of the prior (habitual) intention during the experiment in Fig. 3b and compare the maximal STD in the re-adaptation stage with different cases (Fig. 3f). It can be seen that the prior intention is more uncertain (with larger variance) from stage 3

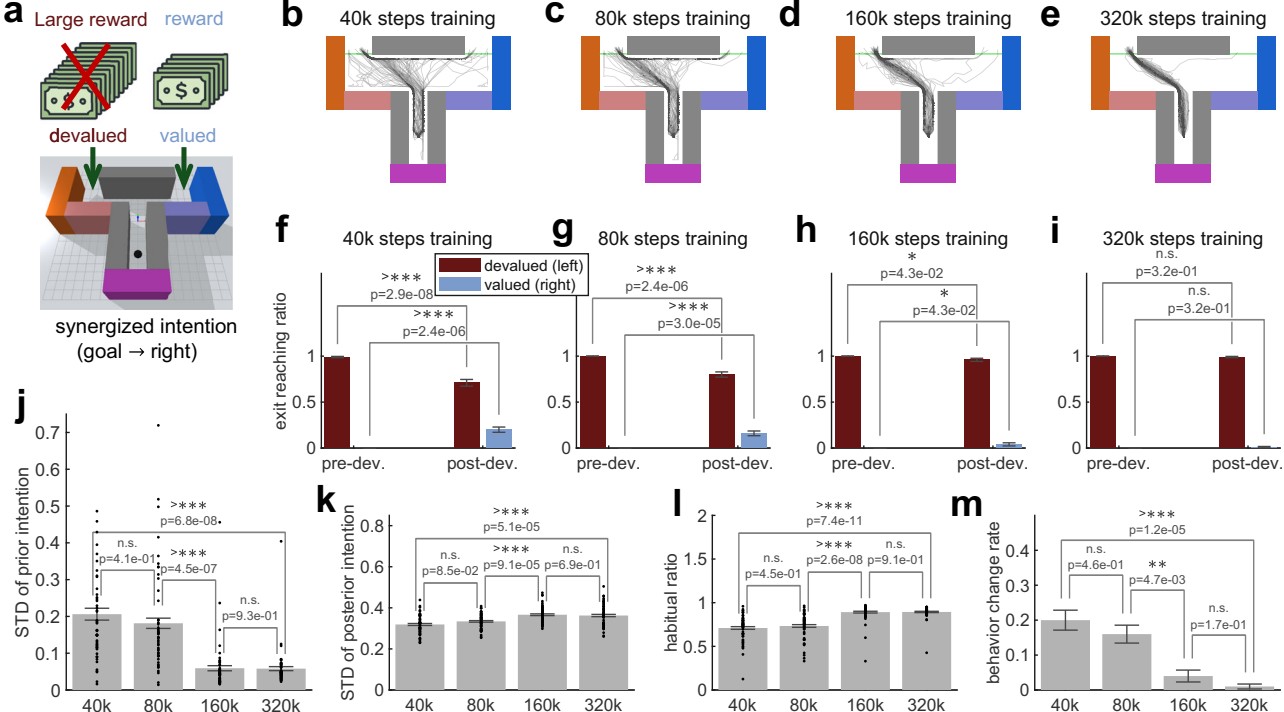

**Fig. 4 | Short-term effect of outcome devaluation. a** Illustration of the experiment. **b–e** Moving trajectories of the agents (with different random seeds) in the first 2 trials after devaluation of the left exit. Each subpanel corresponds to a case of different training steps before devaluation. **f–i** Bar plot of the reaching count for the devalued (left) and valued (right) exits in the last 2 trials before devaluation (pre-dev.) and in the first 2 trials after devaluation (post-dev.). Two-sided $\chi^2$ test is used ($n = 100$: 50 random seeds × 2 trials). Statistics of the mean values of (**j**)

standard deviation of the prior intention (**k**) standard deviation of the posterior intention and (**l**) habitual ratio in the first 2 trials after devaluation. Two-sided Welch's $t$-test is used is used ($n = 50$ random seeds). **m** Change in the ratio of reaching the right exit before and after devaluation. The results show that behavior change from devalued exit is less likely with over-training. Two-sided $\chi^2$ test is used ($n = 100$: 50 random seeds × 2 trials). In **f–m**, data are presented as mean values ± standard error of the mean.

onward, if stage 2 includes more steps. A larger uncertainty about habitual behavior forces the agent to rely more on goal-directed behavior (Fig. 3c, g). Importantly, the goal-directed system also takes more time to re-adapt itself to the new reward configuration with more training in stage 2, which can be reflected from the result that the free energy loss is larger in the case of more training in stage 2 (Fig. 3d, h), potentially due to the KL-divergence regularization between habitual and goal-directed intentions. Our simulation results provides a computational explanation for the long-term resistance of behavior after extensive training: when rewarding outcomes change, the agent becomes more uncertain about its habitual behavior, and it relies more on the goal-directed intention. However, the goal-directed system also needs time to re-adapt, which also partly accounts for the behavior resistance. This is slightly different from the conventional view that the resistance of behavior is mainly owed to a refractory habit[3], yielding more neuroscientific experiments to comprehensively address the underlying mechanisms.

## Immediate behavior change after devaluation

The last section investigated the long-term effect of outcome devaluation by showing the dynamical process of change in the agent's behavior during continuous learning (long-term effect). Meanwhile, a well-recognized experimental approach to measure whether the behavior is habitual or goal-directed is to observe how the agent behaves immediately after devaluation[6,45,52]. We can make an analogy in simulations by analyzing the agent's behavior in the very first trials in stage 3, i.e., about the short-term effect of outcome devaluation. Our simulated experiments mimic the process of outcome devaluation such as food aversion by poisoning or over-eating[6,45] by switching the agent's goal observation from that at the left exit to

that at the right exit at the beginning of stage 3 in the habitization experiment (Fig. 2a), which reflects a decrease of the value of going to the left exit.

We conducted simulated experiments in four cases (40k, 80k, 160k, 320k steps in stage 2), and Fig. 4 analyzes the short-term effect of outcome devaluation. As demonstrated in Fig. 4b–m, our simulation results are consistent with the experimental finding[6,45,52]: An agent with fewer training steps before devaluation shows a lower habitual ratio (i.e., more goal-directed) after devaluation (Fig. 4l). Therefore, agents with 40k or 80k steps training significantly change their behavior, following devaluation, from reaching the left exit to reaching the right exit (Fig. 4f, g), but such behavior change is less obvious with extensive training (160k and 320k steps, Fig. 4h, i).

An interesting observation from Fig. 4c–e is that the agent sometimes first approaches the left exit but realize that it should go to the right later, similar to what people may do. Figure 4m reveals a decline in behavior change toward the valued exit with increased training, suggesting that an over-trained agent struggles to shift from their established preferences. This is due to a significant increase in the precision of prior intention with over-training (Fig. 4j). Although the variance of the posterior intention also varies (Fig. 4k), we can observe a higher habitual ratio after devaluation with over-training (Fig. 4l).

To briefly conclude, our simulation results provide computational modeling of both long-term (Fig. 3) and short-term (Fig. 4) effects of outcome devaluation. We show that after devaluation, an agent with moderate training steps can change its behavior immediately and re-learn the optimal policy quickly as its intention is more goal-directed; whereas an agent with extensive training shows a persistence in its original behavior since its intention is more habitual.

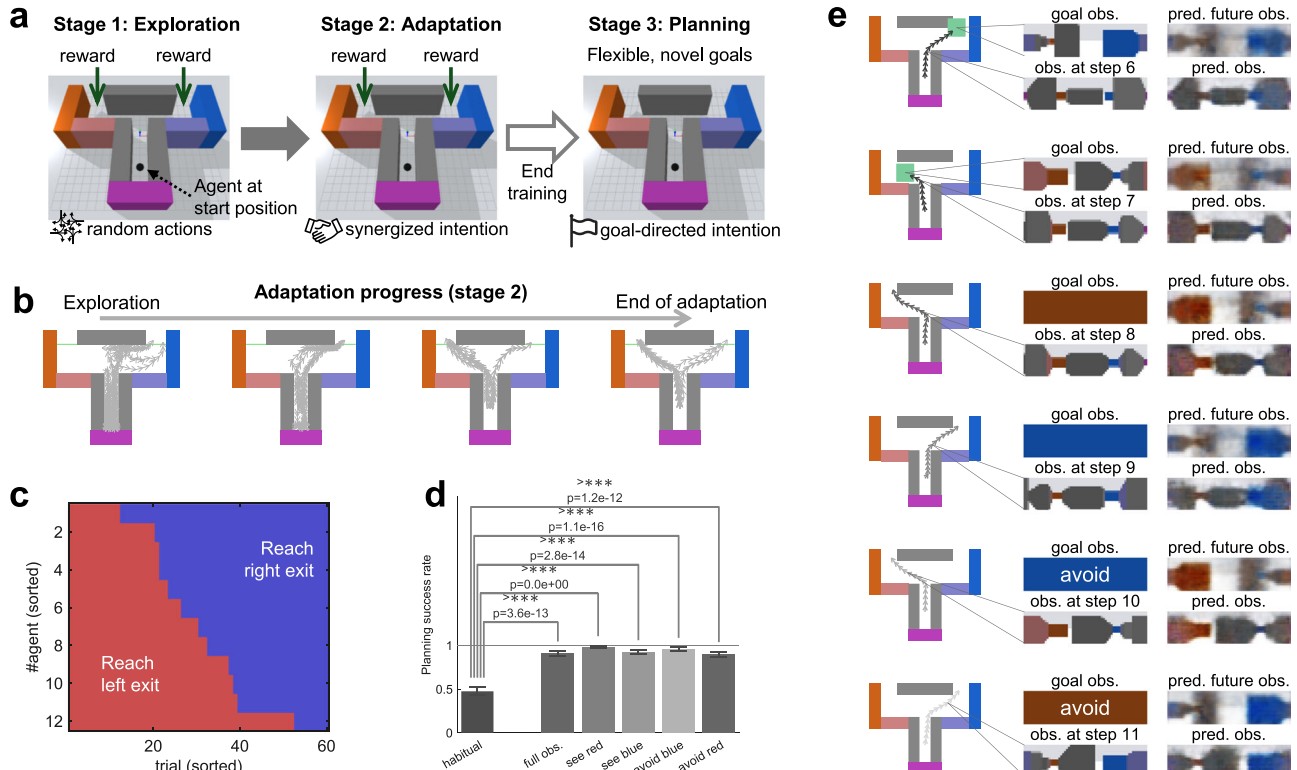

**Fig. 5 | The trained agent with diverse habitual behavior can flexibly perform goal-directed planning. a** Illustration of the experimental setting. Unlike the previous habitization experiment, the rewards are the same for the left and right exits. After stage 2 (adaptation), the model is fixed, and we test the agent's goal-directed planning capacity (stage 3). **b** An example agent behavior (movement trajectories of 10 trials in each plot, aerial view) during stage 2. **c** Statistics of policy diversity using purely habitual behavior (actions computed by prior intention). Totally 12 agents, trained with different random seeds, are tested for 60 trials for each. **d** Statistics of success rate in planning (tested using 12 agents and 10 episodes for each agent in each case) with different kinds of goals: habitual means using purely habitual behavior even though the goal is given, which serves as a baseline.

full obs. means that the goal is provided as a full observation from the goal position in the maze. The color goals refer to objectives involving the presence of more or less amounts of a specified color in future observations (see Methods). Data are presented as mean values ± standard error of the mean ($n = 120$). Two-sided $\chi^2$ test is used. **e** Examples of movement trajectories and internal predictions of current and future observations in goal-directed planning. Full observation from goal position is provided to the agent as goal in the top two examples (the green squares indicate goal regions). The bottom four cases correspond to the color goals. The agent performs success goal-directed behavior for goal position at an exit, at an intermediate position of the maze, as well as color goals, with reasonable sensory predictions.

## Flexible goal-directed planning for novel goals

In previous sections, we have considered the automatic transition from goal-directed to habitual behavior with repetitive training to reduce computational burdens. On the other hand, a relatively less explored problem is how the agent performs goal-directed behavior for novel, untrained task goals[2]. Intelligent biological agents are known to be capable of solving various tasks in a zero-shot manner. For example, a skilled painter can draw a rabbit with mouse ears even if he or she has not done so before. However, this problem is challenging and non-trivial for embodied AI[44,53]. Here, we argue that the connection with habitual intention by the complexity term is crucial for effective goal-directed planning for novel goals. The synergy between two behaviors is the key, explained below. The agent must use its internal model about the environment to perform a mental imaginary to search for a sequence of actions to fulfill the goal (i.e., model-based planning by action search). However, the search space in real-world tasks with continuous actions is unacceptably large (e.g., there are practically infinite ways to buy a battery). Instead of searching for all possible actions, as in conventional model-based action search[54], our framework constrains the search space for goal-directed actions (posterior) with habits (prior). The KL-divergence (complexity) term used in AIf acts as a regulator to keep the goal-directed intention close to the (reward-maximizing) habitual one. For example, if you know a supermarket on the way from work to home (habitual behavior), buying a battery there will save mental effort. Of course, there is no free lunch;

goal-directed planning should require the habitual behavior to be abundant and diverse enough to include action experiences to fulfill the goal. This is also true for humans. For example, one cannot create a detailed oil painting like that of the Mona Lisa without extensive painting experience.

We conducted another simulated experiment (Fig. 5a) to demonstrate the planning capacity using the Bayesian behavior framework. The experiment includes three stages, where the first stage is for exploration using random actions (no learning) as in the previous experiment. The agent performs sensorimotor learning in the second stage, which is similar to the adaptation stage in the previous experiment. However, the rewards at left and right exits are the same so that the agent may acquire diverse habitual behaviors (with probability to either left or right). The agent is trained by RL and free energy minimization using the same algorithm and hyperparameters as in the previous experiments. Since the rewards in the left and right exits are equal, the agent randomly samples the goal observation at each episode with an equal chance of being from the observation at the left or right exits. Then, in the third stage, the agent stops learning (model not updated) and is tested how well it can accomplish novel task goals.

Figure 5b illustrates the development of one example agent's behavior (aerial view) during stage 2 (more results about the learned internal representation in the model can be found in Supplementary Fig. 5). The agent acquires diverse (randomly escaping from top left or top right) and effective (using few steps to finish and avoiding hitting

the wall) behaviors, using the synergized intention which is partly directed by the goal observation. We are also curious about the case without goal specification, i.e., the learned habitual behavior based solely on prior intention $z_t^p$. Figure 5c visualizes the diversity of habitual behavior of the trained agents at the end of stage 2, showing that an agent does not stick to any one exit side, thanks to the stochastic nature of the intention. The diverse experiences also help the agent to acquire an internal predictive model in the goal-directed system for visual observations in most places of the T-maze environment.

Our framework enables the agent to achieve a goal that is not explicitly involved during RL, i.e. zero-shot transfer from reward-maximizing, habitual control to reward-free, goal-directed planning. Here, goal-directed behavior is generated by AIf to compute a $z_t^q$ that minimizes the corresponding free energy, i.e., reducing the gap between the predicted future outcome and the goal while minimizing complexity (Eq. (11)). Then, $z_t^q$ is used to compute the action using the shared policy network. To demonstrate the flexibility of goal-directed behavior using our framework, we performed experiments with providing three kinds of goals to the agent (see Methods for details): (1) the full observation at a target position is provided as the goal observation (full obs. in Fig. 5d); (2) observing a color in the future as much as possible (see red and see blue in Fig. 5d); (3) avoiding observing a color in the future (avoid blue and avoid red in Fig. 5d). The agents perform goal-directed behavior with a high success rate (Fig. 5d), where a goal-directed trial is considered successful if the agent reaches the target region when the goal is full observation, or it reaches the corresponding exit if the goal involves a color.

Figure 5e provides more detailed examples of goal-directed planning. It can also be seen that in cases where the goal involves observing more or less of the red or blue color, the agent also makes reasonably realistic future predictions, while it tries to fulfill the goal. The goal-directed behavior in these cases is approximately in the distribution of habitual behavior, given that the learned habitual behavior contains the diverse options of escaping from either the left exit or the right one. This elucidates one of the most important concepts in our framework – goal-directed planning is constrained and helped by habits (via the KL-divergence minimization between $z_t^q$ and $z_t^p$). For example, if a person wants to move a mug from one place to another, usually the person's habitual gesture to hold the mug will be used instead of other gestures that may also hold the mug. We argue that this should be an essential property for efficient goal-directed planning because the habits already shaped greatly reduce the search space of possible actions for the goal[55,56]. Thus, goal-directed active inference in our framework conducts a search for the relatively more reward-maximizing actions that may lead to future observations close to the goal. This property is crucial to address the efficiency-flexibility dilemma.

## Discussion

Efficient (cost-saving) and flexible (generalizable to novel goals) behavior is crucial for biological agents. We proposed the Bayesian behavior framework, suggesting a theoretical paradigm to consider habitual and goal-directed behavior. Beyond existing computational models[5,15,21,22] considering discrete observations and actions, our framework, empowered by variational Bayes in deep learning, enables simulations using vision-based tasks with continuous motor action, thus reducing the distance from the behavior of real animals[57]. Our framework sheds light on how to best achieve both efficiency and flexibility. Recall the three questions raised at the beginning of this article; now we provide our answers to them after having demonstrated our simulation results.

How does an agent arbitrate between model-free, habitual behavior and model-based, goal-directed one? We assume that the agent leverages the synergized intention given by the inverse variance-weighted average of the two intentions. Although the idea of

uncertainty-based competition between the two behaviors has been discussed in previous studies[5,6,15], they did not reveal the fundamental principle for computing uncertainty for the habitual or goal-directed system. Our formulation of habitual and goal-directed intentions as Bayesian variables provides a natural and direct measurement of the uncertainty of behavior as the statistical variance of the intention distribution, which changes autonomously and dynamically during training by minimizing the free energy and the RL loss.

How does an agent autonomously transfer from slow, goal-directed, to fast, habitual behavior with repetitive trials? We show the dynamic change of the variance of intention (uncertainty) in the continuous learning process that simulates the extinction of goal-directed behavior (stage 2 in Fig. 2) and the extinction of habitual behavior (beginning stage 3 in Fig. 2). The variance of the prior, habitual intention is at first large when adapting to a new task, and it then gradually becomes smaller with repetitive trials (compared with posterior variance) due to the simplicity of model-free decision. This explains why the balance gradually leans toward habitual intention with extensive training. In particular, we introduce a mechanism to early-stop goal-directed AIf when the synergized intention is precise enough. In this way, the agent saves energy in model-based computation while maintaining high behavior precision. Since the prior intention becomes sufficiently accurate after abundant training, the goal-directed AIf needs fewer iterations to make the variance of synergized intention small enough (Fig. 2h), thus explaining a faster computation for decision making with the course of training. Furthermore, we have investigated the agent's behavior after a decrease of the value of the originally best outcome, similar to the outcome devaluation paradigm that is commonly used in experimental research[6,45]. We show results that align with experimental findings and discuss the underlying mechanisms. After devaluation, a moderately trained agent can immediately change its behavior as its intention is more goal-directed, whereas an over-trained agent persists in its old behavior due to a higher habitual ratio.

How does an agent perform goal-directed planning for a novel goal that has not been trained to accomplish? Our answer is that the agent should have an internal predictive model of the environment; thus, it can perform a mental search of motor patterns to achieve the goal. Importantly, it is too costly to search all possible motor actions. Instead, goal-directed intention should be inferred with a constraint from habitual intention (by the KL-divergence term in AIf). In this way, goal-directed planning is more effective, since well-developed low-level motor skills have already been formed in the habitual intention and in the shared policy network. Our thoughts suggest a paradigm to elucidate the generalizability of human behavior to novel goals.

Furthermore, our work also challenges the conventional AI approaches for goal-conditioned decision making[58], which treats the goal as an input to the model and outputs the goal-directed action. Such a treatment has two crucial limitations. First, the goal needs to be explicitly involved in training, in which the agent is rewarded when it achieves the given goal (see Supplementary Related Work). Second, the full state of the goal needs to be given at each trial. For example, if the desired goal is to observe more red colors, it is unknown how to provide the input. Simply using an all-red image as input may not be appropriate since such an observation is impossible. On the contrary, our framework, which is based on PC, considers the goal by using the variational free energy as a loss function to minimize (Eq. (11)). The free energy with respect to the goal is much more flexible. Also, consider the case where the desired goal is to observe more red colors; we can simply replace the prediction error term in the free energy (Eq. (11)) with a loss function reflecting how red the predicted image is.

It should be noted that some parts and sub-ideas of our work have been proposed in the literature, to which we refer the detailed discussion in the Supplementary Related Work. However, our unique contribution is the realization of a unified deep learning framework

that synergizes (1) active inference and reinforcement learning; (2) habitual and goal-directed behaviors; (3) model-based planning and model-free control; (4) free energy minimization and reward maximization, fused via variational Bayes. The fusion of distinct computational principles into one framework contributes to the cost-benefit trade-off[19,27] by the complementary usage of each part. Our work may also serve as a computational platform for modeling behavior in neurological disorders[19,59], of which extensive discussion is deferred to the Supplementary Discussion.

The Bayesian behavior framework has certain limitations. We mainly focus on proof of concept using a fundamental T-maze task, which is nevertheless challenging due to the high-dimensional nature of the first-person vision used for observation. We have yet to tackle more complex motor control tasks, which are commonly seen in real animal behavior[60]. Another limitation is that the agent may not be able to reach states or places that are not covered by its habitual behavior in goal-directed planning. This scenario is relatively rare, but it may occur (e.g., an intentional loss to a weaker opponent in a sport game). To overcome this limitation, it may be necessary to conduct additional learning or search for raw actions instead of relying on the learned motor skills, which can be more time-consuming and resource-intensive.

Regarding future research, the intrinsic generation of goals appears to be an important direction to answer the ultimate questions of how autonomous agents can self-develop[61]. Another important aspect to address is the hierarchical nature of predictive coding[31,62], which should be reflected in the model structure. Moreover, a significant improvement may come from integrating more modalities such as natural language and sound. The format of the goal could be much more flexible by introducing pre-trained multimodality models[63], which provide extensive possibilities for incorporating different sensory modalities into goal-directed intention.

## Methods
### Main network structures
Our model leverages the variational Bayesian method, which is commonly used in deep learning[36,42,64]. The core of our model is a 4-dimensional latent variable $z_t$, also referred to as intention in

this paper.

A visualized diagram can be checked in Fig. 1. The contextual cue (here visual observation) $\bar{x}_t$ (we use the symbol $\bar{x}$ to indicate the ground truth of observations) is first processed by a convolutional neural network (CNN) (Table 1) followed by a recurrent neural network (RNN), in particular, a one-layer gated recurrent unit (GRU)[65]. At environment step $t$, the GRU takes $\bar{x}_t$ (processed by CNN) as input and encodes contextual information into the RNN state $h_t$.

The core of our framework, the latent intention $z_t$, should be paid attention to. $z_t$ is a Bayesian variable that can be sampled with either its prior or posterior distribution. The policy network may take the prior intention $z_t^p$ as input and give out the habitual action; take the posterior $z_t^q$ as input and give out the goal-directed action; or take the synergistic intention $z_t^s$ as input. In other words, habitual and goal-directed behaviors use the same policy network, which allows motor skill sharing[3]. In our implementation, the policy network is a 2-layer multi-layer perceptron (MLP).

The posterior intention $z_t^q$ is computed differently in learning and behaving, which will be explained in the following sections. To predict the current and future observation from $z_t$, a 2-layer MLP followed by a de-convolutional neural network (specified in Table 2) is used as the image decoder (Fig. 1a), whose input is the latent intention $z_t^q$. Unless specified, the width of each hidden layer and the GRU is 256 and the activation functions in fully connected layers are the ReLU function.

### Computing prior intention
The prior distribution of intention $z_t^p$, which determines habitual behavior, should depend on contextual cues[3]. Thus, in our model, $z_t^p$ is always (in learning and behaving) computed using an MLP whose input is $h_t$. $z_t^p$ follows diagonal Gaussian distribution $\mathcal{N}(\mu_t^p, \sigma_t^p)$, where

$$
\begin{aligned}
\mu_t^p &= \mathrm{MLP}(h_t), \\
\xi_t^p &= \mathrm{MLP}(h_t), \\
\sigma_t^p &= \mathrm{softplus}(\xi_t^p) = \ln(1 + e^{\xi_t^p}).
\end{aligned}
\tag{6}
$$

### Learning
During learning (Fig. 1b), the posterior $z_t^q$ follows diagonal Gaussian distribution $\mathcal{N}(\mu_t^q, \sigma_t^q)$ and is estimated by amortized inference[36] using the corresponding contemporary and future observations $\bar{x}_t, \bar{x}_{t'}$ (since the hindsight observations in the future are available in experience replay),

$$
\begin{aligned}
\mu_t^q &= \mathrm{MLP}(\phi(\bar{x}_t, \bar{x}_{t'})), \\
\xi_t^q &= \mathrm{MLP}(\phi(\bar{x}_t, \bar{x}_{t'})), \\
\sigma_t^q &= \mathrm{softplus}(\xi_t^q) = \ln(1 + e^{\xi_t^q}),
\end{aligned}
\tag{7}
$$

where $t'$ is uniformly sampled from range $[t+1, t_{\mathrm{episodeend}}]$, and $\phi$ is a convolutional neural network specified in Table 1.

### Table 1 | Structure of the convolutional neural network

| Layer | Type | Kernel Size | Stride | Padding | Channels | Activation |
|-------|------|-------------|--------|---------|----------|------------|
| 1 | Conv2d | (4, 4) | (2, 2) | (1, 1) | 8 | ReLU |
| 2 | Conv2d | (4, 4) | (2, 2) | (1, 1) | 16 | ReLU |
| 3 | Conv2d | (4, 4) | (2, 2) | (1, 1) | 16 | ReLU |
| 4 | Conv2d | (2, 4) | (2, 2) | (0, 1) | 64 | ReLU |
| 5 | Conv2d | (1, 4) | (1, 4) | (0, 0) | 256 | ReLU |
| 6 | Flatten | – | – | – | – | – |

The input is the RBG array of an image.

### Table 2 | Structure of the de-convolutional neural network

| Layer | Type | Kernel Size | Stride | Padding | Channels | Activation |
|-------|------|-------------|--------|---------|----------|------------|
| 1 | ConvTranspose2d | (1, 4) | (1, 1) | (0, 0) | 64 | ReLU |
| 2 | ConvTranspose2d | (2, 4) | (1, 2) | (0, 1) | 16 | ReLU |
| 3 | ConvTranspose2d | (4, 4) | (2,2) | (1, 1) | 16 | ReLU |
| 4 | ConvTranspose2d | (4, 4) | (2, 2) | (1, 1) | 8 | ReLU |
| 5 | ConvTranspose2d | (3, 3) | (2, 2) | (1, 1) | 8 | ReLU |
| 6 | Conv2d | (3, 3) | (1, 1) | (1, 1) | 6 | – |

The final layer has 6 channels, of which the first half is the prediction of RBG channels of the current observation $x_t$ and the second half corresponds to future observation $x_{t'}$.

As in typical deep RL with experience replay[66], the neural network models of the agent are updated using stochastic gradient descent every $N$ environment steps, where $N = 5$ in our case. At each update, a batch (60 sequences of length 60) of sensorimotor experience (including reward signals) is randomly sampled from the replay buffer, and all networks are trained in one gradient step in an end-to-end manner using the following loss function (here $t$ denotes the step in the recorded sequence, and the loss is averaged over the whole batch):

$$\mathcal{L} = \underbrace{\underbrace{-\mathbb{E}_{q(z_t|\bar{x}_t,\bar{x}_{t'})}\left[\ln P(\boldsymbol{x}_t,\boldsymbol{x}_{t'}=\bar{\boldsymbol{x}}_t,\bar{\boldsymbol{x}}_{t'}|\boldsymbol{z}_t)\right]}_{\text{posterior prediction error}} + \underbrace{\beta_z D_{\text{KL}}\left[q(\boldsymbol{z}_t|\bar{\boldsymbol{x}}_t,\bar{\boldsymbol{x}}_{t'})||p(\boldsymbol{z}_t|\boldsymbol{h}_t)\right]}_{\text{complexity}}}_{\text{free energy}}$$
$$+ \underbrace{\beta_a\mathbb{E}_{q(z_t|\bar{x}_t,\bar{x}_{t'})}\left[\mathcal{L}_{\text{policy}}\right] + \mathcal{L}_{\text{value}}(\bar{\boldsymbol{x}}_t)}_{\text{RL loss function}},$$

(8)

where $\ln P(\boldsymbol{x}_t,\boldsymbol{x}_{t'}=\bar{\boldsymbol{x}}_t,\bar{\boldsymbol{x}}_{t'}|\boldsymbol{z}_t^q) = \ln P(\boldsymbol{x}_t=\bar{\boldsymbol{x}}_t|\boldsymbol{z}_t^q) + \ln P(\boldsymbol{x}_{t'}=\bar{\boldsymbol{x}}_{t'}|\boldsymbol{z}_t^q)$ is the joint likelihood of correct prediction for current and future observations (the bar symbol indicates the ground truth recorded in experience replay). The coefficients $\beta_z, \beta_a$ are the coefficients that determine the balance among these terms. We set $\beta_z = 0.1$ and $\beta_a = 10^5$. The term $\mathbb{E}_{q(z)}\left[\mathcal{L}_{\text{policy}}\right]$ is the loss function of policy learning conditioned on posterior $\boldsymbol{z}$.

Note that although the policy loss is expected over the posterior distribution $q(z)$, it also enhances the performance of habitual actions (using the prior distribution of z) if considering the complexity term together (see Methods). We employ one of the most popular base RL algorithms, the soft actor-critic (SAC)[40,40]. In actor-critic algorithms, value functions, which estimate long-term cumulative rewards over a policy, need also to be learned. We use value networks independent of the main model to learn the value functions of SAC (the last term $\mathcal{L}_{\text{value}}(\bar{\boldsymbol{x}}_t)$). Each value network is a 2-layer MLP. Note that the input to each value network is the original observation encoded by a convolutional neural network (CNN, Table 1), thus the value network is independent of the main network and will only be involved in training[67]. The hyperparameters of SAC are selected following the original work[40] except that we set the target entropy to be linearly reduced from 0 to -4 in the learning course to realize motor noise annealing.

The log-likelihood term in Eq. (8) ($-\mathbb{E}_{q(z_t|\bar{x}_t,\bar{x}_{t'})}\left[\ln P(\boldsymbol{x}_t,\boldsymbol{x}_{t'}=\bar{\boldsymbol{x}}_t,\bar{\boldsymbol{x}}_{t'}|\boldsymbol{z}_t)\right]$) is the posterior prediction errors of the current and future observations $\boldsymbol{x}_t,\boldsymbol{x}_{t'}$ (Fig. 1b). In our work, as image observations are considered, we model each pixel value (ranged in [0, 1]) as the probability of a Bernoulli distribution independent of other pixels as in the variational auto-endoer (VAE) paper[36]. Therefore, the expectation of the log-likelihood $\mathbb{E}_{q(z)}\ln P(\boldsymbol{x}_t=\bar{\boldsymbol{x}}_t)$ can be computed as analytic form (and similarly for $\mathbb{E}_{q(z)}\ln P(\boldsymbol{x}_{t'}=\bar{\boldsymbol{x}}_{t'})$):

$$\mathbb{E}_{q(z)}\ln P(\boldsymbol{x}_t=\bar{\boldsymbol{x}}_t|\boldsymbol{z}_t) = \mathbb{E}_{q(z)}[\bar{\boldsymbol{x}}_t\ln\boldsymbol{x}_t(\boldsymbol{z}_t) + (1-\bar{\boldsymbol{x}}_t)\ln(1-\boldsymbol{x}_t(\boldsymbol{z}_t))].$$

(9)

The KL-divergence term $D_{\text{KL}}\left[q(\boldsymbol{z}_t) \| p(\boldsymbol{z}_t)\right]$ can be analytically given as both the prior and posterior follow diagonal Gaussian distribution[36]:

$$D_{\text{KL}}\left[q(\boldsymbol{z}_t) \| p(\boldsymbol{z}_t)\right] = \ln\frac{\boldsymbol{\sigma}_t^p}{\boldsymbol{\sigma}_t^q} + \frac{(\boldsymbol{\mu}_t^q - \boldsymbol{\mu}_t^p)^2 + (\boldsymbol{\sigma}_t^q)^2}{2(\boldsymbol{\sigma}_t^p)^2} - \frac{1}{2}$$

(10)

### Behaving

We consider a typical episodic RL setting[66], which is a reasonable description of how an animal explores a new environment and learns by trial and error. At each environment step $t$, the agent interacts with the environment by computing motor action $\boldsymbol{a}_t$ using synergistic latent

intention $\boldsymbol{z}_t^s$ (inverse variance weighted average of $\boldsymbol{z}_t^q$ and $\boldsymbol{z}_t^p$). More specifically, the policy network computes a stochastic policy parameterized by $\boldsymbol{\mu}_t^a = \text{MLP}(\boldsymbol{z}_t^s)$ and $\boldsymbol{\sigma}_t^a = \text{MLP}(\boldsymbol{z}_t^s)$, and $\boldsymbol{a}_t$ is given by $\boldsymbol{a}_t = \tanh(\boldsymbol{\mu}_t^a + \epsilon_t\boldsymbol{\sigma}_t^a)$, where ∘ is the Hadamard (element-wise) production and $\epsilon_t$ follows diagonal unit Gaussian distribution. For better exploration, $\epsilon_t$ is given by pink noise as suggested in a recent work[68]. In the goal-directed planning stage in the second experiment (Fig. 5d, e), the agent uses the posterior intention $\boldsymbol{z}_t^q$ instead of $\boldsymbol{z}_t^s$ to perform purely goal-directed behavior. See the following sections for the detailed computing process of $\boldsymbol{z}_t^q$.

The agent receives a scalar reward $r_t$ at each step. If the agent reaches any of the exits, it will receive a reward of 1000 (large reward in the first experiment and in the second experiment) or 500 (small reward in the first experiment). If the agent has a collision with the wall, it will get a small negative reward of −1. Otherwise, the reward is 0. The environment also provides a termination signal $\text{done}_t$, and the episode (trial) is reset when $\text{done}_t = \text{True}$. The agent stores its experience $(\bar{\boldsymbol{x}}_t,\bar{\boldsymbol{x}}_{t+1},\boldsymbol{a}_t,r_t,\text{done}_t)$ in a replay buffer for experience replay in training[69] after each step. The replay buffer can store up to $2^{13}$ sequences of length up to 60. The oldest experience will be replaced by a new one when the replay buffer is full.

### Goal-directed planning by active inference

Here, we detail the active inference[20] process to compute the posterior, goal-directed intention $\boldsymbol{z}_t^q$. We fix the model weights and bias while treating $(\boldsymbol{\mu}_t^q,\boldsymbol{\sigma}_t^q)$ as optimized variables (in practice, $\mathcal{E}^q$ is the variable to be optimized instead of $\boldsymbol{\sigma}^q$ like in Eq. (7)), and optimize them to minimize the variational free energy loss function w.r.t. the goal observation $\boldsymbol{x}_g$ (the current step is denoted by $t$ and the actual current observation is $\bar{\boldsymbol{x}}_t$):

$$\mathcal{L}^{\text{AIf}} = \underbrace{-\ln P(\boldsymbol{x}_t,\boldsymbol{x}_{t'}=\bar{\boldsymbol{x}}_t,\boldsymbol{x}_g|\boldsymbol{z}_t^q)}_{\text{prediction errors}} + \underbrace{\beta_z D_{\text{KL}}\left[q(\boldsymbol{z}_t)||p(\boldsymbol{z}_t|\boldsymbol{h}_t)\right]}_{\text{complexity constrain}},$$

(11)

where $\ln P(\boldsymbol{x}_t,\boldsymbol{x}_{t'}=\bar{\boldsymbol{x}}_t,\boldsymbol{x}_g|\boldsymbol{z}_t^q) = \ln P(\boldsymbol{x}_t=\bar{\boldsymbol{x}}_t|\boldsymbol{z}_t^q) + \ln P(\boldsymbol{x}_{t'}=\boldsymbol{x}_g|\boldsymbol{z}_t^q)$ is the joint likelihood of correct prediction for current observation and goal accomplishment in future observations. In each trial in the habitization experiment (Figs. 2 and 3), the goal observation $\boldsymbol{x}_g$ is randomly sampled from the agent's memory where the agent solved the task and got the largest rewards (the goal selection mechanism is beyond our scope).

We use an evolutionary strategy to optimize the intention $\boldsymbol{z}_t^q$ that minimized $\mathcal{L}^{\text{AIf}}$. In particular, we employed the cross-entropy method[42] to infer $(\boldsymbol{\mu}_t^q,\boldsymbol{\sigma}_t^q)$. The population is 256, and the top 32 are retained for the next generation. During inference, if the mean of $\boldsymbol{\sigma}_t^s$ is smaller than $\sigma^{\text{thres}}$ 0.05 (the agent is certain enough about its intention), the evolutionary process is stopped to save time and energy (the early stop mechanism in Fig. 1d). The maximum number of generations is 16.

### Planning for novel goals

Supposing that the agent has learned diverse habitual behavior (e.g., the agent may move with different routes), our framework allows it to perform zero-shot goal-directed planning for a goal not encountered in training (e.g., provided by human user).

Predictive coding and AIf makes this kind of zero-shot goal-directed planning possible, as our model predicts future observations with the intention $\boldsymbol{z}_t^q$. Notably, the goal here can be of highly flexible choice (Fig. 5). The most basic case is where the agent is provided with its visual image as a goal. In this case, the observation prediction error term can be computed in the same way as training (Eq. (9)). The goal can also be part of the image by masking the other parts in the prediction error. Furthermore, the goal may also be a specific color, and the prediction error is computed as the difference between predictive future observation and an image full of such a color. In practice, suppose that the goal color is $G$ (RGB value); For the case that the goal is to pursue more

red (or blue) colors in future observations, we replace the prediction error in Eq. (11) with $\sum_{i,j,c} \mathbb{1}(|\boldsymbol{x}_{i,j,c}^{\text{pred}} - G_c| < 0.25)$, scaled by 5 times, where $\boldsymbol{x}_{i,j,c}$ is the predicted pixel value at the the $i$th row, $j$th column and color channel $c$, and the operator $\mathbb{1}$ returns 1 if the inequality is satisfied otherwise returns 0. Inversely, we may minimize the negative of this difference so that the goal is to observe less of this color by replacing the prediction error with $-\sum_{i,j,c} \mathbb{1}(|\boldsymbol{x}_{i,j,c}^{\text{pred}} - G_c| < 0.25)$, scaled by 100 times. The scaling factors were heuristically hand-tuned. In principle, any goal that can be reflected by a loss function about the observation prediction can be considered.

### On the complexity term in free energy

Here we quickly show that the KL-divergence between posterior and prior intentions enables a synergy of habitual and goal-directed behaviors: while the model optimizes the performance of motor actions using posterior goal-directed intention (Fig. 1b), the motor actions using prior habitual intention is also being optimized. Consider that we want to maximize the logarithm of the expected likelihood of the action computed from prior $\boldsymbol{z}_t^p$ equal to the optimal action $\boldsymbol{a}_t^*$ (assuming that the optimal action can be estimated using the learned value function). The loss function can be written as:

$$
\begin{aligned}
\mathcal{L}_{\text{habitual behavior policy}} &:= -\ln \mathbb{E}_{p(\boldsymbol{z}_t|\boldsymbol{h}_t)}\left[P(\boldsymbol{a}(\boldsymbol{z}_t) = \boldsymbol{a}_t^*)\right] \\
&= -\ln \int_{\boldsymbol{z}_t} P(\boldsymbol{a} = \boldsymbol{a}_t^*|\boldsymbol{z}_t) p(\boldsymbol{z}_t|\boldsymbol{h}_t) d\boldsymbol{z}_t \\
&= -\ln \int_{\boldsymbol{z}_t} \frac{q(\boldsymbol{z}_t|\bar{\boldsymbol{x}}_t, \bar{\boldsymbol{x}}_{t'})}{q(\boldsymbol{z}_t|\bar{\boldsymbol{x}}_t, \bar{\boldsymbol{x}}_{t'})} P(\boldsymbol{a} = \boldsymbol{a}_t^*|\boldsymbol{z}_t) p(\boldsymbol{z}_t|\boldsymbol{h}_t) d\boldsymbol{z}_t.
\end{aligned}
$$

By Jensen's inequality, we have

$$
\begin{aligned}
\mathcal{L}_{\text{habitual behavior policy}} &\le -\int_{\boldsymbol{z}_t} q(\boldsymbol{z}_t|\bar{\boldsymbol{x}}_t, \bar{\boldsymbol{x}}_{t'}) \ln \frac{P(\boldsymbol{a}_t = \boldsymbol{a}_t^*|\boldsymbol{z}_t) p(\boldsymbol{z}_t|\boldsymbol{h}_t)}{q(\boldsymbol{z}_t|\bar{\boldsymbol{x}}_t, \bar{\boldsymbol{x}}_{t'})} d\boldsymbol{z}_t \\
&= -\int_{\boldsymbol{z}_t}\left[ q(\boldsymbol{z}_t|\bar{\boldsymbol{x}}_t, \bar{\boldsymbol{x}}_{t'}) \ln P(\boldsymbol{a}_t = \boldsymbol{a}_t^*|\boldsymbol{z}_t) - q(\boldsymbol{z}_t|\bar{\boldsymbol{x}}_t, \bar{\boldsymbol{x}}_{t'}) \ln \frac{q(\boldsymbol{z}_t|\bar{\boldsymbol{x}}_t, \bar{\boldsymbol{x}}_{t'})}{p(\boldsymbol{z}_t|\boldsymbol{h}_t)} \right] d\boldsymbol{z}_t \\
&= \underbrace{-\mathbb{E}_{q(\boldsymbol{z}_t|\bar{\boldsymbol{x}}_t, \bar{\boldsymbol{x}}_{t'})}\left[\ln P(\boldsymbol{a}_t = \boldsymbol{a}_t^*|\boldsymbol{z}_t)\right]}_{\text{posterior policy loss}} + \underbrace{D_{KL}\left[q(\boldsymbol{z}_t|\bar{\boldsymbol{x}}_t, \bar{\boldsymbol{x}}_{t'}) \| p(\boldsymbol{z}_t|\boldsymbol{h}_t)\right]}_{\text{complexity}}.
\end{aligned}
$$

$$(12)$$

This derivation is similar to the derivation of variational lower bound[36]. Interestingly, the complexity term is the same as the KL-divegence term in the free energy. Equation (12) explains that why Eq. (8) is actually optimizing habitual behavior (given by prior $\boldsymbol{z}_t^p$) despite that the policy loss term is expected over posterior $\boldsymbol{z}_t^q$: Eq. (12) is the upper bound of the policy loss of habitual behavior. Therefore, by training to decrease the upper bound, the policy loss of habitual behavior becomes lower, making habitual behavior more effective (gaining more rewards). This can be intuitively understood as learning with posterior intention as oracle information to guide habitual behaviors[70].

### Convolutional and de-convolutional networks structure

The structures of the convolutional and de-convolutional neural networks used for image encoding and decoding in this work are specified in Tables 1 and 2.

### Reporting summary

Further information on research design is available in the Nature Portfolio Reporting Summary linked to this article.

## Data availability

The data used in this study were generated by computer simulations by our code. All the code-generated data used in this study have been deposited in Zenodo (https://doi.org/10.5281/zenodo.10987473).

## Code availability

We wrote the simulation code by ourselves. The source code is available at https://github.com/oist-cnru/The-Bayesian-Behavior-Framework. For

data analysis, we used superbar (https://github.com/scottclowe/superbar) for the bar plot, and ciplot (https://www.mathworks.com/matlabcentral/fileexchange/63314-ciplot-lower-upper-x-colour-alpha) for the confidence interval region for curves. Otherwise we used our custom data analysis code (see https://github.com/oist-cnru/The-Bayesian-Behavior-Framework for the source code and reproduction guideline).

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

## Acknowledgements

This work was supported by Okinawa Institute of Science and Technology (OIST) and Microsoft Research. Kenji Doya was also supported by the Japan Society for the Promotion of Science KAKENHI Grant Numbers JP16K21738, JP16H06563, and JP23H04975, and the Japan Agency for Medical Research and Development (AMED) Grand Number JP23dm0207001 and JP23dm0307009. The authors would like to thank the members of the cognitive neurorobotics research unit, the neural computation unit in OIST, and MSRA shanghai; as well as Sang Wan Lee, Zhaoyun Chen, and Tadashi Kozuno for insightful discussions and comments.

## Author contributions

D.H. wrote the computer code, performed the simulations and analyzed the results. J.T. conceived the initial idea. D.H and J.T together shaped the detailed framework. D.H., K.D., D.L. and J.T. were involved in the discussions to improve the model and task design. All the four authors participated in the paper writing.

## Competing interests

The authors declare no competing interests.
