## [Peer Review File · Nature Communications]

REVIEWER COMMENTS

Reviewer #1 (Remarks to the Author):

The manuscript entitled 'HABITS AND GOALS IN SYNERGY: A VARIATIONAL BAYESIAN FRAMEWORK FOR BEHAVIOR' addresses the question how to formulate a mathematical model that comprises, based on a single mechanism, both goal-directed and habitual behavior. The authors propose to use a variational framework using a latent variable called intention. In this model, habitual actions corresponds to actions that are computed just based on the prior of the intention variable, while goal-directed actions are based on the posterior of the latent variable. The authors illustrate the principle based on a T-maze task.

In summary, the paper is a potentially interesting contribution to the question how habitual and goal-directed behavior can be computed by a single system. This is important because currently, the two seemingly different types of behavior are often understood as two distinct systems in the cognitive neuroscience and psychology literature.

Major comments:

The introduction is written with a specific focus that seems at times confusing for someone from the cognitive neuroscience and psychology community actually working on the question how habitual behaviour can be explained in a model. For example, the authors formulate three 'critical questions in cognitive neuroscience'. I am not so sure that these three questions are currently such critical questions in cognitive neuroscience. For example, in the first question, what do the authors mean by 'diverse while effective habitual behavior'? isn't the point of habitual behavior that it is non-diverse but rather automatic in the sense that it is called upon when a specific stimulus set is sensed? Or the third question seems to be important in a rather global sense but it is not clear why it is a question in this specific paper whether agents can reach goals that have not been trained yet.

It is also confusing why the authors do not seem to cite reviews or similar that comprise the latest insights about habits from the psychology or cognitive neuroscience literature. For example, Wood & Runger 2016 is a standard citation in this sense. This gives the manuscript the air of a detachment from actual experimental research into habits and goal-directed actions.

This is probably the explanation why the authors forgo to include key questions in the experimental habit literature. For example, the point about habits are that they are fast. Would this be predicted by the present model? Also, habits usually become faster and more inflexible the longer the training is. Is this the case with the model? These and other questions are central to the current understanding of habitual behavior in cognitive neuroscience and should be addressed by a model, or at least discussed at all.

Another weak point, which the author try to address, is the experimental standard concept that goal-directed behavior precedes the development of habitual behavior. This is quite intuitive and seems to contradict the main idea of the model. The authors add simulations to show that it is possible to model this, under some additional assumptions. Although a laudable attempt, it is not compelling that such a fundamental concept is only added as an apparent after-thought, where the main idea introduced over 200+ lines of manuscript is not in line with standard experimental concepts and understanding.

It is also an open question that the authors cite active inference papers but do not address at all how these techniques have been used before to address the question of the authors, i.e. how habitual behaviour can be explained in a variational Bayesian framework. To my knowledge, there are two contributions of interest. The first, Friston et al 2016, the authors actually cite but do not relate or discuss that paper's approach how to model habits. The second, is Schwöbel et al. (2021), where the authors also use active inference to integrate habitual and goal-directed behavior.

In summary, as much as I applaud the authors to come up with an innovative approach to solve a problem, the manuscript reads like an attempt to bridge the gap between habitual and goal-directed behavior without too much relying on experimental facts, findings and concepts. This seems to me a severe weakness because a modelling attempt should relate to the current experimental insight and findings to make the contribution accessible to an important part of the target audience.

Minor points:

Line 19 'Habitual behavior is typically model-free (MF),...': this seems a categorical error: behavior cannot be a model. One is the observed phenomenon, the other is a model. I guess the authors mean: 'Habitual behavior is typically modelled by'

Line 22: Same for: 'Goal-directed behavior is model-based (MB),...'

Line 36: Same thing: what does '... habitual (MF)...' mean? Do the authors assume the equality of habitual behaviour and actions as computed by model-free reinforcement learning? This would be a strong assumption.

Line 72: category error: predictive coding is not a key neural substrate, it's a conceptual or theoretical framework, as also stated in the next sentence.

Lines 90 to 92: This statement doesn't seem correct and is misleading. There have been attempts to address this, see major comments above.

Lines 93/94: isn't it one of the core topics of active inference,

which is presented here as the implementation of predictive coding, that it implements flexibility and goal-directed planning? Why would this be new or worth mentioning? Maybe formulate more precisely what is meant as novelty?

Fig 1 is not clear at all. For example, add explanation about pyramid and camel.

Reviewer #2 (Remarks to the Author):

Review of HABITS AND GOALS IN SYNERGY: A VARIATIONAL BAYESIAN FRAMEWORK FOR BEHAVIOR

This submission to Nature Communications describes a computational model that is intended to account for the distinction between goal-directed and habitual control. The key innovation of this model is that it is intended to accommodate both goal-directed and habitual action selection within the same overlapping architecture, and in which actions learned in a habitual way within the model can be repurposed to be utilized in a goal-directed manner and vice versa, thereby avoiding the need for them to be trained separately. The authors use a deep-learning approach, and the distinction between goal-directed and habitual learning is based on whether the prior distribution of intention without a goal representation is used to guide action-selection, or whether a posterior distribution is used to drive behavior, in which the objective of the agent is to minimize free energy of the goal representation and goal observation.

This is an interesting model of action-selection that marries free energy, RL and deep-learning into a unified framework for action-selection. I wonder how well this model would perform in more complex "real-world" tasks beyond the rather simplified T-maze.

However, while this a potentially interesting model architecture, it is unclear how or why it is relevant for understanding the brain. The neurobiological relevance of this approach is not currently obvious, and the

extent to which this model can capture patterns of actual animal behavior and accommodate what is known about the underlying neurobiology of goal-directed and habitual control is unclear.

(1) From the outset it is very unclear what the objective of the authors actually is with respect to this model. Is it to produce a model that can succeed in implementing goal-oriented behavior and habits in a unified architecture in a way that is not related to the actual biology and animal behavior? That is are the authors trying to come up with a convenient computational solution to this problem without any particular reference to what is known about animal behavior or neurobiology? Or are the authors actually trying to come up with a model that explains how animals might try to solve this problem, constrained by what is known about animal behavior and the underlying neurobiology? It would be important for the authors to clarify this point. If the former, I think this model would need to be tested against a much wider variety of potential tasks to prove its capacity beyond the very simple T-maze task that the authors present and would need to be compared against other models as benchmarks, which is not what the authors currently present. So, I'm going to assume that the authors are intending to present this model as being relevant to actual biology and behavior, and I will proceed with the review on this assumption.

(2) The authors appear to be focused on two main computational problems, the first is the need to implement goal-directed and habitual control systems within a unified architecture. However, why is this a constraint that needs to be satisfied? What is the empirical evidence for this common architecture between goal-directed and habitual control that the authors are trying to satisfy? To my understanding there is clear evidence for at least partly dissociable neural substrates for goal-directed and habitual control. As a result, it seems to me the authors are trying to implement an architecture that doesn't actually reflect a property of the underlying neurobiology. The other problem the authors allude to needing to solve is the need to have generalization / transfer of learning from goal-directed to habitual actions within the same representations. However, once again, there is no evidence that this a problem the brain needs to solve, as given what we know, it is entirely plausible that learning of goal-directed and habitual actions proceeds in parallel but distinct systems.

(3) On the other hand, there are actual problems that real brains appear to be able to solve that are not currently well understood such as how the brain transfers control from goals to habits with overtraining, or other manipulations such as the use of a variable interval schedule vs a ratio schedule etc, which the authors do not attempt to capture in their model.

(4) More seriously than the above, the authors conceptualization of habits and goals are somewhat limited and don't match onto how these processes are being studied in the real biological and psychological literature. First of all, a key feature of goal-directed control as it is studied in the animal literature is outcome value sensitivity – that is the animal performs goal-directed actions in a manner

that reflects the value of the current goal, and changes its behavior after changes in goal-value. This might be trivially implemented in their model by suggested different objective representations for goals as a function of changing value, but the model appears to miss a central part of the goal-directed architecture altogether which is how goals are valued, and how a model could actively keep track of changing values of goals, in order to change behavior. In other words, the model doesn't select its own goals based on outcome-values, these goals are imposed on the system. The current task that the authors use of the T-maze doesn't capture this notion of devaluation of outcomes, and thus it doesn't show how this current model can implement goal-directed action selection in a manner that relates to the conceptualization of goal-directed behavior in the literature. Moreover, it is unclear to what extent the behavior the authors are modeling as "habitual" corresponds to what is known as habitual behavior in the literature in terms of devaluation insensitive behavior (though I suspect this model when in habit mode would behave in such a devaluation insensitive way, given it is essentially relying on model-free RL).

(5) It would also be useful to compare and contrast the author's proposed model against other architectures that have been proposed, such as for instance Friston's et al (2016) framework for the distinction between goals and habits, that also uses the free energy principle.

When taken together, while the authors present an interesting architecture – it is unclear how relevant what the authors have found is to how these behaviors are implemented in a real brain, unclear why this architecture is desirable as a means of accounting for how real brains might solve this task, and unclear how this model captures the array of behaviors found when comparing goal-directed and habitual action selection in real animals.

Response to reviewer comments for paper “Synergizing habits and goals with variational Bayes” (previous named “habits and goals in synergy: a variational Bayesian framework for behavior”)

By authors of manuscript NCOMMS-23-26556-T

(The original reviewer comments are attached at the end of this document)

We are grateful for the thorough and insightful feedback provided by the reviewers and editors. We are deeply encouraged by the comments from both reviewers that our framework is innovative and interesting. Crucially, the issues highlighted by the reviewers have been instrumental in driving substantial enhancements to our work. We have significantly improved our study with a wealth of new results to address the concerns. We first summarize the major changes from the previous version, followed by point-to-point responses.

- (1) The title has been changed to “Synergizing Habits and Goals with Variational Bayes” to be more concise and to comply with the journal’s requirement of no colon in the title.
- (2) We have improved the deep learning model structure to be simpler while remaining as effective as before. However, our main idea has not changed: treating habit as a prior latent intention and goal-direction as a posterior one, as well as learning and using them in synergy by constraining their divergence.
- (3) We propose an autonomous mechanism to arbitrate between habitual and goal-directed intentions based on the uncertainty (variance) of them (see the section “Learning and behaving under the Bayesian behavior framework” in Results).
- (4) We have conducted abundant simulations explaining the automatic transition from goal-directed behavior to habit, addressing a major concern from both reviewers. This is now a central result in our work (“Automatically transition from goal-directed to habitual behavior” and “More training makes behavior more resistant” sections in Results).
- (5) We have rewritten most parts of the main texts to comply with these changes. The current manuscript should be easier to follow for psychologists, neuroscientists, and a general audience, while keeping a detailed, precise description of our algorithm and model in the Methods section for AI researchers.

Detailed responses:

Reviewer #1

>>The introduction is written with a specific focus that seems at times confusing for someone from the cognitive neuroscience and psychology community actually working on the question how habitual behaviour can be explained in a model. For example, the authors formulate three 'critical questions in cognitive neuroscience'. I am not so sure that these three questions are currently such critical questions in cognitive neuroscience. For example, in the first question, what do the authors mean by 'diverse while effective habitual behavior'? isn't the point of habitual behavior that it is non-diverse but rather automatic in the sense that it is called upon when a specific stimulus set is sensed? Or the third question seems to be important in a rather global sense but it is not clear why it is a question in this specific paper whether agents can reach goals that have not been trained yet.

We agree with the reviewer and have significantly revised the manuscript. The revised paper focuses on three pivotal questions in line with current neuroscientific and psychological research: 1 . How does an agent arbitrate between model-free, habitual behavior and mode-based, goal-directed one? 2 . How does an agent autonomously transfer from slow, goal-directed to fast, habitual behavior with repetitive trials? 3 . How does an agent perform goal-directed planning for a novel goal that has not been trained to accomplish? Our revised paper first uses the proposed Bayesian behavior framework to computationally address questions 1 and 2 (new Results in the revision), and then turns to question 3 (main result in the initial submission) as an extension from questions 1 and 2.

>>It is also confusing why the authors do not seem to cite reviews or similar that comprise the latest insights about habits from the psychology or cognitive neuroscience literature. For example, Wood & Runger 2016 is a standard citation in this sense. This gives the manuscript the air of a detachment from actual experimental research into habits and goal-directed actions.

This is probably the explanation why the authors forgo to include key questions in the experimental habit literature. For example, the point about habits are that they are fast. Would this be predicted by the present model? Also, habits usually become faster and more inflexible the longer the training is. Is this the case with the model? These and other questions are central to the current understanding of habitual behavior in cognitive neuroscience and should be addressed by a model, or at least discussed at all.

We appreciate this constructive comment. The revised manuscript now refers to Wood & Runger 2016, and more importantly, has significantly improved the model and increased simulation experiments to align with actual experimental research (see the Results regarding the new Figures 2 and 3).

In particular, we address the detailed questions from the reviewer under our framework. First, habits are faster and computationally cheaper (Figure 1 c,d). Second, decision-making becomes faster (taking fewer outcome-related computations) with training (Figure 2 h). Third, behavior becomes more resistant to change with longer training (Figure 3). We believe that our study, in its current form, provides theoretical

insights into these questions that are central to the current understanding of habitual behavior in cognitive neuroscience.

>>Another weak point, which the author try to address, is the experimental standard concept that goal-directed behavior precedes the development of habitual behavior. This is quite intuitive and seems to contradict the main idea of the model. The authors add simulations to show that it is possible to model this, under some additional assumptions. Although a laudable attempt, it is not compelling that such a fundamental concept is only added as an apparent after-thought, where the main idea introduced over 200+ lines of manuscript is not in line with standard experimental concepts and understanding.

As responded above, our revised manuscript now focuses on the arbitration between two systems, and habit automaticity, while keeping the main results in the initial submission in the last section of results. The current results should have resolved this concern.

>>It is also an open question that the authors cite active inference papers but do not address at all how these techniques have been used before to address the question of the authors, i.e. how habitual behaviour can be explained in a variational Bayesian framework. To my knowledge, there are two contributions of interest. The first, Friston et al 2016, the authors actually cite but do not relate or discuss that paper's approach how to model habits. The second, is Schwöbel et al. (2021), where the authors also use active inference to integrate habitual and goal-directed behavior.

We agree that Friston et al. (2016) and Schwöbel et al. (2021) have discussed similar topics and are closely related to ours. However, our approach is fundamentally different from theirs. Some key differences are:

- They modeled the learning and interplay of the two kinds of behaviors under Active Inference (Alf) theory, while we show how (model-free) Reinforcement Learning (RL) and Alf work in synergy, as RL is also well-supported by neuroscience.
- They considered discrete states and actions as well as known world models, whereas our work simulates sensorimotor tasks with visual sensation and continuous motor actions where the agent learns everything from scratch by self-exploration.
- We demonstrate our framework's capacity to perform flexible goal-directed planning for unseen goals, which were not covered in their work.

We thank the reviewer for raising this question. We updated the manuscript to refer to these two studies in the introduction, and we provide a detailed discussion about the relation to them in Supplementary Related Work.

>> In summary, as much as I applaud the authors to come up with an innovative approach to solve a problem, the manuscript reads like an attempt to bridge the gap between habitual and goal-directed

behavior without too much relying on experimental facts, findings and concepts. This seems to me a severe weakness because a modelling attempt should relate to the current experimental insight and findings to make the contribution accessible to an important part of the target audience.

We appreciate the reviewer for providing this critical perspective to improve our research. We believe our revised manuscript, with abundant new simulation results and significant text revisions to comply with neuroscientific findings, should address the concerns.

Minor points:

>>Line 19 'Habitual behavior is typically model-free (MF),...': this seems a categorical error: behavior cannot be a model. One is the observed phenomenon, the other is a model. I guess the authors mean: Habitual behavior is typically modelled by'

Line 22: Same for: 'Goal-directed behavior is model-based (MB),...'

Thanks, we modified the sentences to fix the categorical errors.

>>Line 36: Same thing: what does '... habitual (MF)...' mean? Do the authors assume the equality of habitual behaviour and actions as computed by model-free reinforcement learning? This would be a strong assumption.

Thanks for pointing it out. Our work does not simply assume habitual behavior and actions are merely computed by model-free reinforcement learning. We have changed it to "habitual and goal-directed behavior have been treated as two independent problems..."

>> Line 72: category error: predictive coding is not a key neural substrate, it's a conceptual or theoretical framework, as also stated in the next sentence.

Lines 90 to 92: This statement doesn't seem correct and is misleading. There have been attempts to address this, see major comments above.

Lines 93/94: isn't it one of the core topics of active inference, which is presented here as the implementation of predictive coding, that it implements flexibility and goal-directed planning? Why would this be new or worth mentioning? Maybe formulate more precisely what is meant as novelty?

The reviewer is correct. We have rewritten the introduction and removed these descriptions.

>>Fig 1 is not clear at all. For example, add explanation about pyramid and camel.

We have re-plotted Figure 1 to provide a clearer and more intuitive explanation of our updated framework.

Reviewer #2

>> *This is an interesting model of action-selection that marries free energy, RL and deep-learning into a unified framework for action-selection. I wonder how well this model would perform in more complex "real-world" tasks beyond the rather simplified T-maze.*

However, while this a potentially interesting model architecture, it is unclear how or why it is relevant for understanding the brain. The neurobiological relevance of this approach is not currently obvious, and the extent to which this model can capture patterns of actual animal behavior and accommodate what is known about the underlying neurobiology of goal-directed and habitual control is unclear.

We are glad that the reviewer finds our model interesting. We have updated our model, added new simulation results, and reorganized the manuscript. The revised manuscript should now better align with relevant neurobiological concepts. Please also refer to our response for the following detailed comments.

Scaling up to more complex "real-world" tasks will indeed be of significant value to investigate. However, since our work currently focuses on modeling the interaction between habitual and goal-directed behavior in line with psychological and neuroscientific facts, the T-maze environment is ideal for demonstrating our ideas with the added simulation experiments in the revision.

Nonetheless, this T-maze environment takes a step from existing computational studies addressing similar topics [a-d] using categorical (discrete options) of states and actions. We consider a sensorimotor task with vision-based sensory and continuous motor action, and each episode (trial) contains 10+ steps, providing much larger freedom of movements. We consider this work as the foundation for future work in more challenging tasks, which is indeed our ongoing work.

[a] Lee S W, Shimojo S, O'Doherty J P. Neural computations underlying arbitration between model-based and model-free learning[J]. *Neuron*, 2014, 81(3): 687-699.

[b] Friston K, FitzGerald T, Rigoli F, et al. Active inference and learning[J]. *Neuroscience & Biobehavioral Reviews*, 2016, 68: 862-879.

[c] Kim D, Park G Y, O' Doherty J P, et al. Task complexity interacts with state-space uncertainty in the arbitration between model-based and model-free learning[J]. *Nature communications*, 2019, 10(1): 5738.

[d] Schwöbel S, Marković D, Smolka M N, et al. Balancing control: a Bayesian interpretation of habitual and goal-directed behavior[J]. *Journal of mathematical psychology*, 2021, 100: 102472.

>> *(1) From the outset it is very unclear what the objective of the authors actually is with respect to this model. Is it to produce a model that can succeed in implementing goal-oriented behavior and habits in a unified architecture in a way that is not related to the actual biology and animal behavior? That is are the authors trying to come up with a convenient computational solution to this problem without any particular*

reference to what is known about animal behavior or neurobiology? Or are the authors actually trying to come up with a model that explains how animals might try to solve this problem, constrained by what is known about animal behavior and the underlying neurobiology? It would be important for the authors to clarify this point. If the former, I think this model would need to be tested against a much wider variety of potential tasks to prove its capacity beyond the very simple T-maze task that the authors present and would need to be compared against other models as benchmarks, which is not what the authors currently present. So, I'm going to assume that the authors are intending to present this model as being relevant to actual biology and behavior, and I will proceed with the review on this assumption.

We appreciate the reviewer for raising the question about the central objective of our study. We have updated our model, added new simulation results, and re-organized the manuscript to better address the biological relevance. We added comprehensive simulations to use our framework to explain the automatic transition from goal-directed to habitual behavior, followed by our original results about flexible goal-directed planning for novel goals. We believe that our revised manuscript, with these improvements and re-emphasis of the research objective, provides novel theoretical insights into the corresponding neuroscientific problems.

>>(2) The authors appear to be focused on two main computational problems, the first is the need to implement goal-directed and habitual control systems within a unified architecture. However, why is this a constraint that needs to be satisfied? What is the empirical evidence for this common architecture between goal-directed and habitual control that the authors are trying to satisfy? To my understanding there is clear evidence for at least partly dissociable neural substrates for goal-directed and habitual control. As a result, it seems to me the authors are trying to implement an architecture that doesn't actually reflect a property of the underlying neurobiology. The other problem the authors allude to needing to solve is the need to have generalization / transfer of learning from goal-directed to habitual actions within the same representations. However, once again, there is no evidence that this a problem the brain needs to solve, as given what we know, it is entirely plausible that learning of goal-directed and habitual actions proceeds in parallel but distinct systems.

We believe there is a misunderstanding of our core idea, likely due to our initial lack of clarity. Our core idea is not that goal-directed and habitual systems should be unified into one. Rather, our framework centers on the interaction between them. We model the intentions for habit and goal-orientation as two Bayesian latent variables, which can be interpreted as prior and posterior in variational Bayesian theory. The two intentions, while computable from distinct systems in the brain, interact with each other in learning and behaving through a KL-divergence regularization between them (see Figure 1 b,c in the revised manuscript). Another key idea is that both habitual and goal-directed behaviors use the same policy network (mapping intention to detailed motor actions) thus the motor skills are shared. This is consistent with neuroscientific recognition about common motor output pathways like M1 and the brainstem [e].

[e] Redgrave P, Rodriguez M, Smith Y, et al. Goal-directed and habitual control in the basal ganglia: implications for Parkinson's disease[J]. Nature Reviews Neuroscience, 2010, 11(11): 760-772.

>>(3) *On the other hand, there are actual problems that real brains appear to be able to solve that are not currently well understood such as how the brain transfers control from goals to habits with overtraining, or other manipulations such as the use of a variable interval schedule vs a ratio schedule etc, which the authors do not attempt to capture in their model.*

We have added simulations using our framework to address the transfer from goals to habits with overtraining (Figure 2), and the effect of varying training durations (Figure 3). These results are now in the revised manuscript. The investigation about a variable interval schedule vs a ratio schedule [f] is also interesting, which we are considering to incorporate into future work.

[f] Yamada K, Toda K. Habit formation viewed as structural change in the behavioral network[J]. Communications biology, 2023, 6(1): 303.

>>(4) *More seriously than the above, the authors conceptualization of habits and goals are somewhat limited and don't match onto how these processes are being studied in the real biological and psychological literature. First of all, a key feature of goal-directed control as it is studied in the animal literature is outcome value sensitivity – that is the animal performs goal-directed actions in a manner that reflects the value of the current goal, and changes its behavior after changes in goal-value. This might be trivially implemented in their model by suggested different objective representations for goals as a function of changing value, but the model appears to miss a central part of the goal-directed architecture altogether which is how goals are valued, and how a model could actively keep track of changing values of goals, in order to change behavior. In other words, the model doesn't select its own goals based on outcome-values, these goals are imposed on the system. The current task that the authors use of the T-maze doesn't capture this notion of devaluation of outcomes, and thus it doesn't show how this current model can implement goal-directed action selection in a manner that relates to the conceptualization of goal-directed behavior in the literature.*

Regarding how goals are selected, our work at initial submission focused on flexible goal-directed planning for novel goals, thus we first let the agent to learn sensorimotor skills without goal-specification, and then “forcing” an unseen goal by the experimenter to the agent (This part corresponds to the results regarding Figure 4 in the revised paper). This is a relatively more under-explored problem that how does an agent computationally infer the actions to satisfy a novel goal (e.g., painting a rabbit with mouse ears).

However, in the current revised version, we highlight the new results about automatic arbitration and transition between these two kinds of behaviors (regarding Figures 2 and 3 in the revised paper). In accordance with the reviewer's comment that goal-selection should be value-sensitive, we now adopt a simple, autonomous goal-selection mechanism: the agent memorizes the observation at positions with the largest reward and uses it as the goal in behaving. When reward devaluation happens, the agent also changes its goal accordingly. While an in-depth investigation of the mechanism for goal selection is beyond our scope, we think this simple mechanism is consistent with neuroscientific conceptualizations.

>> Moreover, it is unclear to what extent the behavior the authors are modeling as "habitual" corresponds to what is known as habitual behavior in the literature in terms of devaluation insensitive behavior (though I suspect this model when in habit mode would behave in such a devaluation insensitive way, given it is essentially relying on model-free RL).

To address this concern, we conducted a simulation experiment about behavior's insensitivity to reward devaluation. Please refer to the result in the revised manuscript regarding Figures 2 and 3.

>>(5) It would also be useful to compare and contrast the author's proposed model against other architectures that have been proposed, such as for instance Friston's et al (2016) framework for the distinction between goals and habits, that also uses the free energy principle.

We agree and have updated the Introduction to refer to Friston's et al [b], as well as another related work by Schwöbel et al. [d] suggested by reviewer #1. However, our approach is fundamentally different from theirs, some key differences are:

- They modeled the learning and interplay of the two kinds of behaviors under Active Inference (Alf) theory, while we show how (model-free) Reinforcement Learning (RL) and Alf work in synergy, as RL is also well-supported by neuroscience.
- They considered discrete states and actions as well as known world models, whereas our work simulates sensorimotor tasks with visual sensation and continuous motor actions where the agent learns everything from scratch by self-exploration.
- We demonstrate our framework's capacity to perform flexible goal-directed planning for unseen goals, which were not covered in their work.

We also have provided detailed discussion about the relation to them in Supplementary Related Work.

>>When taken together, while the authors present an interesting architecture – it is unclear how relevant what the authors have found is to how these behaviors are implemented in a real brain, unclear why this architecture is desirable as a means of accounting for how real brains might solve this task, and unclear how this model captures the array of behaviors found when comparing goal-directed and habitual action selection in real animals.

We again appreciate the constructive comments from the reviewer. We believe our revised manuscript, with abundant new simulation results and significant text revisions to comply with neuroscientific findings, should resolve the previous weaknesses of our work.

REVIEWER COMMENTS

Reviewer #1 (Remarks to the Author):

The manuscript entitled 'HABITS AND GOALS IN SYNERGY: A VARIATIONAL BAYESIAN FRAMEWORK FOR BEHAVIOR' addresses the question how to formulate a mathematical model that comprises, based on a single mechanism, both goal-directed and habitual behavior. The authors propose to use a variational framework using a latent variable called intention. In this model, habitual actions corresponds to actions that are computed just based on the prior of the intention variable, while goal-directed actions are based on the posterior of the latent variable. The authors illustrate the principle based on a T-maze task.

In summary, the paper is a potentially interesting contribution to the question how habitual and goal-directed behavior can be computed by a single system. This is important because currently, the two seemingly different types of behavior are often understood as two distinct systems in the cognitive neuroscience and psychology literature.

Major comments:

The introduction is written with a specific focus that seems at times confusing for someone from the cognitive neuroscience and psychology community actually working on the question how habitual behaviour can be explained in a model. For example, the authors formulate three 'critical questions in cognitive neuroscience'. I am not so sure that these three questions are currently such critical questions in cognitive neuroscience. For example, in the first question, what do the authors mean by 'diverse while effective habitual behavior'? isn't the point of habitual behavior that it is non-diverse but rather automatic in the sense that it is called upon when a specific stimulus set is sensed? Or the third question seems to be important in a rather global sense but it is not clear why it is a question in this specific paper whether agents can reach goals that have not been trained yet.

It is also confusing why the authors do not seem to cite reviews or similar that comprise the latest insights about habits from the psychology or cognitive neuroscience literature. For example, Wood & Runger 2016 is a standard citation in this sense. This gives the manuscript the air of a detachment from actual experimental research into habits and goal-directed actions. This is probably the explanation why the authors forgo to include key questions in the experimental habit literature. For example, the point about habits are that they are fast. Would this be predicted by the present model? Also, habits usually become faster and more inflexible the longer the training is. Is this the case with the model? These and other questions are central to the current understanding of habitual behavior in cognitive neuroscience and should be addressed by a model, or at least discussed at all.

Another weak point, which the author try to address, is the experimental standard concept that goal-directed behavior precedes the development of habitual behavior. This is quite intuitive and seems to contradict the main idea of the model. The authors add simulations to show that it is possible to model this, under some additional assumptions. Although a laudable attempt, it is not compelling that such a fundamental concept is only added as an apparent after-thought, where the main idea introduced over 200+ lines of manuscript is not in line with standard experimental concepts and understanding.

It is also an open question that the authors cite active inference papers but do not address at all how these techniques have been used before to address the question of the authors, i.e. how habitual behaviour can be explained in a variational Bayesian framework. To my knowledge, there are two contributions of interest. The first, Friston et al 2016, the authors actually cite but do not relate or discuss that paper's approach how to model habits. The second, is Schwöbel et al. (2021), where the authors also use active inference to integrate habitual and goal-directed behavior.

In summary, as much as I applaud the authors to come up with an innovative approach to solve a problem, the manuscript reads like an attempt to bridge the gap between habitual and goal-directed behavior without too much relying on experimental facts, findings and concepts. This seems to me a severe weakness because a modelling attempt should relate to the current experimental insight and findings to make the contribution accessible to an important part of the target audience.

Minor points:

Line 19 'Habitual behavior is typically model-free (MF),...': this seems a categorical error: behavior cannot be a model. One is the observed phenomenon, the other is a model. I guess the authors mean: Habitual behavior is typically modelled by'

Line 22: Same for: 'Goal-directed behavior is model-based (MB),...'

Line 36: Same thing: what does '... habitual (MF)...' mean? Do the authors assume the equality of habitual behaviour and actions as computed by model-free reinforcement learning? This would be a strong assumption.

Line 72: category error: predictive coding is not a key neural substrate, it's a conceptual or theoretical framework, as also stated in the next sentence.

Lines 90 to 92: This statement doesn't seem correct and is misleading. There have been attempts to address this, see major comments above.

Lines 93/94: isn't it one of the core topics of active inference, which is presented here as the implementation of predictive coding, that it implements flexibility and goal-directed planning? Why would this be new or worth mentioning? Maybe formulate more precisely what is meant as novelty?

Fig 1 is not clear at all. For example, add explanation about pyramid and camel.

Reviewer #2 (Remarks to the Author):

Review of HABITS AND GOALS IN SYNERGY: A VARIATIONAL BAYESIAN FRAMEWORK FOR BEHAVIOR

This submission to Nature Communications describes a computational model that is intended to account for the distinction between goal-directed and habitual control. The key innovation of this model is that it is intended to accommodate both goal-directed and habitual action selection within the same overlapping architecture, and in which actions learned in a habitual way within the model can be repurposed to be utilized in a goal-directed manner and vice versa, thereby avoiding the need for them to be trained separately. The authors use a deep-learning approach, and the distinction between goal-directed and habitual learning is based on whether the prior distribution of intention without a goal representation is used to guide action-selection, or whether a posterior distribution is used to drive behavior, in which the objective of the agent is to minimize free energy of the goal representation and goal observation.

This is an interesting model of action-selection that marries free energy, RL and deep-learning into a unified framework for action-selection. I wonder how well this model would perform in more complex "real-world" tasks beyond the rather simplified T-maze.

However, while this a potentially interesting model architecture, it is unclear how or why it is relevant for understanding the brain. The neurobiological relevance of this approach is not currently obvious, and the extent to which this model can capture patterns of actual animal behavior and accommodate what is known about the underlying neurobiology of goal-directed

and habitual control is unclear.

(1) From the outset it is very unclear what the objective of the authors actually is with respect to this model. Is it to produce a model that can succeed in implementing goal-oriented behavior and habits in a unified architecture in a way that is not related to the actual biology and animal behavior? That is are the authors trying to come up with a convenient computational solution to this problem without any particular reference to what is known about animal behavior or neurobiology? Or are the authors actually trying to come up with a model that explains how animals might try to solve this problem, constrained by what is known about animal behavior and the underlying neurobiology? It would be important for the authors to clarify this point. If the former, I think this model would need to be tested against a much wider variety of potential tasks to prove its capacity beyond the very simple T-maze task that the authors present and would need to be compared against other models as benchmarks, which is not what the authors currently present. So, I'm going to assume that the authors are intending to present this model as being relevant to actual biology and behavior, and I will proceed with the review on this assumption.

(2) The authors appear to be focused on two main computational problems, the first is the need to implement goal-directed and habitual control systems within a unified architecture. However, why is this a constraint that needs to be satisfied? What is the empirical evidence for this common architecture between goal-directed and habitual control that the authors are trying to satisfy? To my understanding there is clear evidence for at least partly dissociable neural substrates for goal-directed and habitual control. As a result, it seems to me the authors are trying to implement an architecture that doesn't actually reflect a property of the underlying neurobiology. The other problem the authors allude to needing to solve is the need to have generalization / transfer of learning from goal-directed to habitual actions within the same representations. However, once again, there is no evidence that this a problem the brain needs to solve, as given what we know, it is entirely plausible that learning of goal-directed and habitual actions proceeds in parallel but distinct systems.

(3) On the other hand, there are actual problems that real brains appear to be able to solve that are not currently well understood such as how the brain transfers control from goals to habits with overtraining, or other manipulations such as the use of a variable interval schedule vs a ratio schedule etc, which the authors do not attempt to capture in their model.

(4) More seriously than the above, the authors conceptualization of habits and goals are somewhat limited and don't match onto how these processes are being studied in the real

biological and psychological literature. First of all, a key feature of goal-directed control as it is studied in the animal literature is outcome value sensitivity – that is the animal performs goal-directed actions in a manner that reflects the value of the current goal, and changes its behavior after changes in goal-value. This might be trivially implemented in their model by suggested different objective representations for goals as a function of changing value, but the model appears to miss a central part of the goal-directed architecture altogether which is how goals are valued, and how a model could actively keep track of changing values of goals, in order to change behavior. In other words, the model doesn't select its own goals based on outcome-values, these goals are imposed on the system. The current task that the authors use of the T-maze doesn't capture this notion of devaluation of outcomes, and thus it doesn't show how this current model can implement goal-directed action selection in a manner that relates to the conceptualization of goal-directed behavior in the literature. Moreover, it is unclear to what extent the behavior the authors are modeling as "habitual" corresponds to what is known as habitual behavior in the literature in terms of devaluation insensitive behavior (though I suspect this model when in habit mode would behave in such a devaluation insensitive way, given it is essentially relying on model-free RL).

(5) It would also be useful to compare and contrast the author's proposed model against other architectures that have been proposed, such as for instance Friston's et al (2016) framework for the distinction between goals and habits, that also uses the free energy principle.

When taken together, while the authors present an interesting architecture – it is unclear how relevant what the authors have found is to how these behaviors are implemented in a real brain, unclear why this architecture is desirable as a means of accounting for how real brains might solve this task, and unclear how this model captures the array of behaviors found when comparing goal-directed and habitual action selection in real animals.

REVIEWER COMMENTS

Reviewer #1 (Remarks to the Author):

The authors have addressed all my concerns. The manuscript is still written in a style that is often presumably difficult to grasp by an experimental reader in cognitive neuroscience but I believe modelling habitual vs. goal-directed actions requires new ideas.

Reviewer #2 (Remarks to the Author):

The authors have addressed many of the prior concerns and the manuscript is now much improved. Incorporating arbitration is also an interesting contribution to this model. Demonstrating that there is an increased resistance to behavior change following increased training in their model is interesting, especially the observation that in such cases persistent/perseverative behavior is driven by uncertainty in the goal-directed system due to greater free energy loss therein. The new simulations presented in Figure 4 on the role of prior habit learning on sculpting goal-directed planning for novel goals are interesting.

1. However, for the goal-directed vs habit test, the precise manipulation they are using doesn't entirely match up with the formal test of goal-directed vs habitual control as manifested in the classic reward devaluation procedure. In the simulations as presented, the authors change the value of the reward provided by the left arm of the T-maze to zero. The agent then gets to experience that changed reward amount and gets to adjust its behavior eventually converging on the right arm (which previously had the lower reward amount but now has the largest amount given the left arm value has switched to zero).

However, in the psychology literature on goals and habits, the crucial test of whether behavior is goal-directed or habitual is on the next trial after devaluation has occurred (whereby devaluation is implemented outside of the task context). Think of this as being akin to having reward tokens (e.g red and blue coins) which during initial training are both valuable, and then the value of one of the tokens is set to zero after the devaluation has occurred. Crucially, if the animal is goal-directed following this reinforcer devaluation manipulation, it should decrease responding on the action associated with the devalued outcome on the first trial after devaluation because it can retrieve the current the value of the token and associates that with the goal-directed action. Following over-training it does not do this (if behavior is perfectly habitual) but it will persist on responding to the action leading to the now devalued token. Note this experience on the first trial is distinct from the re-learning that occurs after reexperiencing an outcome that has lower value in the trial context (that they currently simulate),

because in the devaluation test manipulation the outcome has never before been experienced before in the task environment in its devalued state, yet a goal-directed animal knows to stop responding on the action on that very first trial following devaluation because it can separately access the current incentive value of the outcome (that was devalued separately from the task context). This test manipulation is also typically done in extinction (without presenting any outcome) so that even on subsequent trials, a difference between goal-directed and habitual control can be observed until the response / action extinguishes completely because no reward is now available to the animal, but I think just showing what happens on the very first trial after devaluation before the devalued outcome is reexperienced would be fine for simulation purposes.

I think it might be helpful to implement a similar manipulation in their simulations to capture this phenomenon, alongside the existing simulations that they present. As it is they are capturing effects of re-learning about the value of the different arms – which could explain increased perseveration on previously rewarded actions following reversal with overtraining, but such a test is not considered a gold-standard test compared to a formal devaluation procedure.

2. Can the authors clarify whether the overtraining phenomenon they observe in their model simulations in which overtraining produces an increased refractory period following a value change is dependent on the introduction of a precision threshold (early stop mechanism) whereby when the prior precision is sufficiently precise the active inference process stops iterating? If so, how dependent are the simulation results on specific values of this stopping threshold?

Minor comments

There are a number of typos / grammatical errors, but I assume these will be fixed following further proof reading.

However, note that the use of the word “habituation” for the process of habit formation is not correct here – habituation typically refers to the phenomenon of sensory habituation which is a different phenomenon. Instead, the correct term is “habitization”.

Response to reviewer comments for paper “Synergizing habits and goals with variational Bayes”

By authors of manuscript NCOMMS-23-26556-A

We highly appreciate the insightful feedback provided by the reviewers and editors. We have complemented the results and revised the manuscript accordingly (highlighted by blue color in the paper). The revision includes:

- (1) Adding new results about the devaluation experiment as suggested by reviewer #2.
- (2) Clarifying the impact of precision threshold of intention in Supplementary.
- (3) Fixing typos / grammatic errors and correcting the term “habituation” to “habitization”.
- (4) Slightly modifying the illustrations of the experiments to improve clarity.

Reviewer #1

>> The authors have addressed all my concerns. The manuscript is still written in a style that is often presumably difficult to grasp by an experimental reader in cognitive neuroscience but I believe modelling habitual vs. goal-directed actions requires new ideas.

We thank the reviewer for the valuable feedback. Our future work will aim to build connections with experimental research to enable a better synergy of theory and practice.

Reviewer #2

>> The authors have addressed many of the prior concerns and the manuscript is now much improved. Incorporating arbitration is also an interesting contribution to this model. Demonstrating that there is an increased resistance to behavior change following increased training in their model is interesting, especially the observation that in such cases persistent/perseverative behavior is driven by uncertainty in the goal-directed system due to greater free energy loss therein. The new simulations presented in Figure 4 on the role of prior habit learning on sculpting goal-directed planning for novel goals are interesting.

1. However, for the goal-directed vs habit test, the precise manipulation they are using doesn't entirely match up with the formal test of goal-directed vs habitual control as manifested in the classic reward devaluation procedure. In the simulations as presented, the authors change the

value of the reward provided by the left arm of the T-maze to zero. The agent then gets to experience that changed reward amount and gets to adjust its behavior eventually converging on the right arm (which previously had the lower reward amount but now has the largest amount given the left arm value has switched to zero).

However, in the psychology literature on goals and habits, the crucial test of whether behavior is goal-directed or habitual is on the next trial after devaluation has occurred (whereby devaluation is implemented outside of the task context). Think of this as being akin to having reward tokens (e.g red and blue coins) which during initial training are both valuable, and then the value of one of the tokens is set to zero after the devaluation has occurred. Crucially, if the animal is goal-directed following this reinforcer devaluation manipulation, it should decrease responding on the action associated with the devalued outcome on the first trial after devaluation because it can retrieve the current the value of the token and associates that with the goal-directed action. Following over-training it does not do this (if behavior is perfectly habitual) but it will persist on responding to the action leading to the now devalued token. Note this experience on the first trial is distinct from the re-learning that occurs after reexperiencing an outcome that has lower value in the trial context (that they currently simulate), because in the devaluation test manipulation the outcome has never before been experienced before in the task environment in its devalued state, yet a goal-directed animal knows to stop responding on the action on that very first trial following devaluation because it can separately access the current incentive value of the outcome (that was devalued separately from the task context). This test manipulation is also typically done in extinction (without presenting any outcome) so that even on subsequent trials, a difference between goal-directed and habitual control can be observed until the response / action extinguishes completely because no reward is now available to the animal, but I think just showing what happens on the very first trial after devaluation before the devalued outcome is reexperienced would be fine for simulation purposes.

I think it might be helpful to implement a similar manipulation in their simulations to capture this phenomenon, alongside the existing simulations that they present. As it is they are capturing effects of re-learning about the value of the different arms – which could explain increased perseveration on previously rewarded actions following reversal with overtraining, but such a test is not considered a gold-standard test compared to a formal devaluation procedure.

We are grateful for the valuable comments by the reviewer about the lack of standard devaluation experiment. Per the reviewer's suggestion, we have implemented a similar simulated experiment to show what happens on the very first trial after devaluation. The results are consistent with the experimental findings that the agent decreases responding on the action associated with the devalued outcome, but it does not do this with over-training. Corresponding results are discussed in the revised manuscript (Figure 4).

>> 2. Can the authors clarify whether the overtraining phenomenon they observe in their model simulations in which overtraining produces an increased refractory period following a value change is dependent on the introduction of a precision threshold (early stop mechanism) whereby when the prior precision is sufficiently precise the active inference process stops iterating? If so, how dependent are the simulation results on specific values of this stopping threshold?

We have examined much larger (0.2) and smaller (0.01) values of the stopping threshold (0.05 in the main texts), an obvious result is that the number of iterations in active inference increased with a smaller threshold (intention requires to be more precise). However, other main results are not affected. We have mentioned this in the main texts, and the corresponding simulations and analysis have been added to the supplementary results (Supplementary Figures 2,3,4).

>>Minor comments

There are a number of typos / grammatical errors, but I assume these will be fixed following further proof reading.

We have conducted proofreading to fix the errors.

>> However, note that the use of the word “habituation” for the process of habit formation is not correct here – habituation typically refers to the phenomenon of sensory habituation which is a different phenomenon. Instead, the correct term is “habitization”.

Thanks for pointing this out, we have revised the name to “habitization”.

REVIEWERS' COMMENTS

Reviewer #2 (Remarks to the Author):

The authors have addressed my remaining concerns. I have no further comments.

Response to reviewer comments for paper “Synergizing habits and goals with variational Bayes”

By authors of manuscript NCOMMS-23-26556-B

We highly appreciate the insightful feedback provided by the reviewers and editors. We have made the final revision to comply with the editorial policies.

Reviewer #1

>> The authors have addressed all my concerns. The manuscript is still written in a style that is often presumably difficult to grasp by an experimental reader in cognitive neuroscience but I believe modelling habitual vs. goal-directed actions requires new ideas.

We thank the reviewer for the valuable feedback. Our future work will aim to build connections with experimental research to enable a better synergy of theory and practice.

Reviewer #2

>> The authors have addressed my remaining concerns. I have no further comments.

We thank the reviewer for the valuable feedback which largely helps to improve our work.